# Training-free Diffusion Model Adaptation for Variable-Sized Text-to-Image Synthesis

**Zhiyu Jin**[1], **Xuli Shen**[1,2], **Bin Li**[1]*, **Xiangyang Xue**[1]
[1] Shanghai Key Laboratory of Intelligent Information Processing
School of Computer Science, Fudan University
[2] UniDT Technology

## Abstract

Diffusion models (DMs) have recently gained attention with state-of-the-art performance in text-to-image synthesis. Abiding by the tradition in deep learning, DMs are trained and evaluated on the images with fixed sizes. However, users are demanding for various images with specific sizes and various aspect ratio. This paper focuses on adapting text-to-image diffusion models to handle such variety while maintaining visual fidelity. First we observe that, during the synthesis, lower resolution images suffer from incomplete object portrayal, while higher resolution images exhibit repetitively disordered presentation. Next, we establish a statistical relationship indicating that attention entropy changes with token quantity, suggesting that models aggregate spatial information in proportion to image resolution. The subsequent interpretation on our observations is that objects are incompletely depicted due to limited spatial information for low resolutions, while repetitively disorganized presentation arises from redundant spatial information for high resolutions. From this perspective, we propose a scaling factor to alleviate the change of attention entropy and mitigate the defective pattern observed. Extensive experimental results validate the efficacy of the proposed scaling factor, enabling models to achieve better visual effects, image quality, and text alignment. Notably, these improvements are achieved without additional training or fine-tuning techniques.

## 1 Introduction

Diffusion models have emerged as a powerful technique for image synthesis [10, 15, 46], achieving state-of-the-art performance in various applications [12, 28, 14]. Among them, text-to-image diffusion models have garnered significant attention and witnessed a surge in demand [40, 37, 36]. Traditionally, diffusion models have adhered to the typical deep learning approach of training and testing on images with predetermined sizes, which generally leads to high-quality results. They still exhibit a range of visual defects and diverse flaws when confronted with a novel synthesizing resolution (e.g., $512^2$ in training while $224^2$ in testing). However, real-world scenarios often demand the generation of images with diverse sizes and aspect ratios, necessitating models that can handle such variety with minimum loss in visual fidelity. The necessity becomes even more severe in the generation of large models. As the size of models continues to increase, the associated training costs experience a substantial surge, thereby posing financial challenges for average programmers and emerging startups, making it unfeasible for them to train specialized models tailored to their specific needs. Consequently, there is an urgent demand to explore methodologies that facilitate the full utilization of open-sourced models trained on fixed sizes.

In regard to this limitation, our first key observation reveals that most instances of poor performance could be attributed to two prevalent patterns: **incomplete or inadequately represented objects** and

---

*Corresponding author

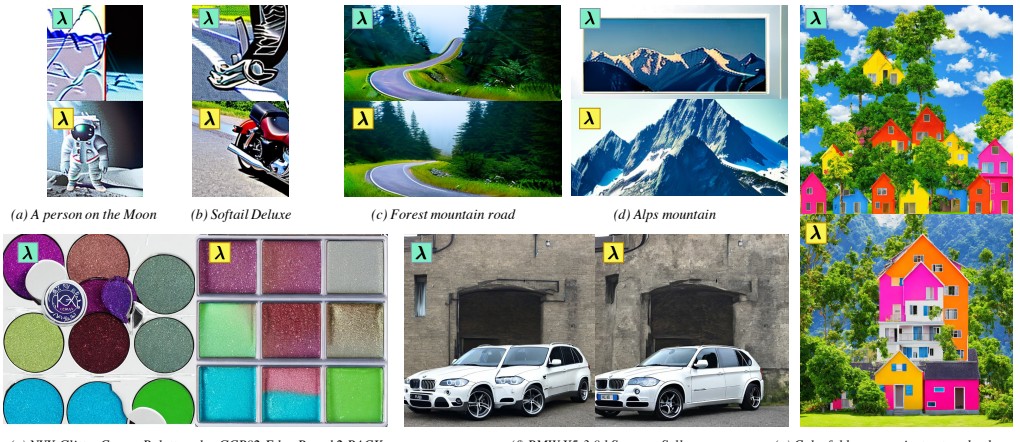

*(a) A person on the Moon*  *(b) Softail Deluxe*  *(c) Forest mountain road*  *(d) Alps mountain*

*(e) NYX Glitter Cream Palette color GCP02 Eden Brand 2 PACK*  *(f) BMW X5 3.0d Swap or Sell*  *(g) Colorful house against nature background*

Figure 1: **Synthesized results** with the proposed scaling factor (marked with yellow icons) and those with original scaling factor (marked with green icons). The method with our scaling factor successfully synthesizes high-fidelity and natural images for different resolutions. Please zoom in for better visual effect.

**repetitively disordered presentations.** We have included several examples of flawed synthesized images in Figure 1, designated by green icons. Note that smaller images (e.g., images (a), (b), (c) and (d)) conform to the first pattern, showcasing inadequately depicted objects. Conversely, larger images (e.g., images (e), (f) and (g)) exhibit the second pattern, generating disordered objects in a repetitive manner. This delineation, also observed in other problematic cases, allows us to formulate our second key observation, where **lower resolution images are more vulnerable to the first pattern, while susceptibility to the second pattern increases in higher resolution images.**

In this work, we tackle the challenge of adapting text-to-image diffusion models to proficiently synthesize images spanning a wide range of sizes, encompassing both low and high resolutions. Our goal is threefold: (1) achieve a higher level of fidelity regardless of synthesized image resolutions; (2) mitigate the abovementioned two patterns; and (3) augment the semantic alignment between the synthesized images and the text prompts.

To accomplish these, our approach centers around the concept of entropy, which measures the spatial granularity of information aggregation. Specifically, when the attention entropy rises, each token is attending to wider spatial information, otherwise the opposite. Bearing that in mind, we establish the statistical relationship between the resolution (the number of tokens) and the entropy of attention maps. Our finding signifies that the attention entropy varies in correspondence with the quantity of tokens, which implies that models are aggregating spatial information in proportion to resolutions. We then establish our key interpretations based on the proportionality. In particular, since narrower attention is paid during the synthesis of low resolution images, models encounter difficulties in representing complete objects with limited spatial information. Conversely, when synthesizing high resolution images, each token tends to conduct a wide spatial aggregation which results in a redundancy of analogous spatial information and the disordered presentations of repeated elements.

Based on the statistical relationship and interpretations, we propose a novel scaling factor specifically designed for mitigating the change of the entropy in visual attention layers. We conduct qualitative and quantitative experiments to validate the efficacy of our scaling factor, which synthesizes results of varied resolutions with better visual effect and achieves better scores in image quality and text alignment. It is worth noting that the improvement is implemented in a training-free manner by replacing a single scaling factor in the attention layers.

The contributions are summarized as follows:

- We observe two distinct patterns of defects that low resolution images are prone to impaired object portrayal while repetitively disordered presentation exhibits in high resolution images.
- We establish a statistical relationship between attention the entropy and the resolution, which interprets the patterns by ascribing them to the lack or the surplus of the spatial information.

- We propose a scaling factor to efficiently improve variable-sized text-to-image synthesis in a training-free manner. We conduct various experiments and analysis to validate the efficacy.

## 2   Related Work

**Image super-resolution and extrapolation.**   An alternative path to generate new images with different resolutions is post-processing synthesized images with image super-resolution and extrapolation methods. Super-resolution methods [18, 6, 53, 54] dedicate to exploiting the high frequency information of images and extending both the height and width of images by a fixed factor, which cannot change the aspect ratio of images. For extrapolation methods, they do have the ability to change the aspect ratio by extending images beyond their boundaries [50, 58, 29]. However, these methods are effective for signals that exhibit an intensive stationary component [5] (e.g., landscapes) while we would like to generate images with various scenes. Additionally, extrapolation methods might have discrepancy with diffusion models, resulting in inharmonious visual effect. Instead of adding more methods for post-processing and making the pipeline lengthy, we seek to synthesize high-fidelity images with diffusion models alone.

**Diffusion models and attention mechanism.**   Diffusion models have emerged as a promising family of deep generative models with outstanding performance in image synthesis [10, 15, 22, 45, 46, 47]. They have significantly revolutionized computer vision research in multiple areas, including image editing [8, 12, 31], inpaintng [28, 39, 56], super-resolution [16, 41] and video synthesis [14, 17]. These models generate high-quality images by smoothly denoising a noisy image sampled from a standard Gaussian distribution through multiple iterations. A key component in diffusion models is the attention module. Previous works such as denoising diffusion probabilistic model [15], denoising diffusion implicit model [46], latent diffusion model [37] have demonstrated the capability empowered by the attention modules. Note that attention modules in diffusion models are dealing with thousands of visual tokens and one of the most well-known research concern of attention is how to lower its conventional quadratic complexity against the large token numbers. To deal with this, most works define a sparse attention pattern to restrict each query to attend to a limited subset of tokens exclusively. The sparse attention pattern could be either defined as fixed windows [23, 52, 49, 38, 30] or learned during training [25, 34, 7, 4, 1, 60, 26, 57, 55, 9, 20]. We draw upon the insights in these works for sparsity controlling and seek to efficiently adapt trained models for variable-sized data.

**Attention entropy.**   Introduced in the information theory by [44], entropy serves as a measure for information and uncertainty. Different research works have introduced entropy to attention maps in various tasks, including semantic segmentation [24], text classification [2], text generation [48], active visual exploration [33] and interpretability analysis [32]. Among them, the work in text generation [48] closely aligns with ours on attention scaling, leveraging Transformers to accommodate sentences of variable lengths. We draw upon the innovations in the entropy of attention maps for information quantification and seek to exploit both of its practicability and interpretability.

## 3   Method

This section establishes the connection between the attention entropy and the token number in a statistical manner and presents a scaling factor replaced into diffusion model for variable-sized image synthesis. Our aim is to alleviate the fluctuation in attention entropy with varying token numbers.

### 3.1   Connection: the attention entropy and the token number

Diffusion models refer to probabilistic models that learn a data distribution by iteratively refining a normally distributed variable, which is equivalent to learn the reverse process of a fixed Markov Chain with a length of $T$. In image synthesis, the most efficacious models make use of a Transformer-based encoder-decoder architecture for the denoising step. This step can be understood as an evenly weighted chain of denoising autoencoders, which are trained to predict a filtered version of their input via cross-attention. Yet the scaling factor inside the attention module of diffusion models is so far an under-explored area of research. In this section, we focus on enhancing conditional diffusion models by adaptively determining the scaling factor of Transformer encoder-decoder backbones.

Let $\mathbf{X} \in \mathbb{R}^{N \times d}$ denotes an image token sequence to an attention module, where $N, d$ are the number of tokens and the token dimension, respectively. We then denote the key, query and value matrices as $\mathbf{K} = \mathbf{X}\mathbf{W}^K, \mathbf{Q}$ and $\mathbf{V}$, where $\mathbf{W}^K \in \mathbb{R}^{d \times d_r}$ is a learnable projection matrix and $d_r$ is the projection dimension. The attention layer ([3, 51]) computes $\text{Attention}(\mathbf{Q}, \mathbf{K}, \mathbf{V}) = \mathbf{A}\mathbf{V}$ with the attention map $\mathbf{A}$ calculated by the row-wise softmax function as follows:

$$\mathbf{A}_{i,j} = \frac{e^{\lambda \mathbf{Q}_i \mathbf{K}_j^\top}}{\sum_{j'=1}^{N} e^{\lambda \mathbf{Q}_i \mathbf{K}_{j'}^\top}}, \tag{1}$$

where $\lambda$ is a scaling factor, usually set as $1/\sqrt{d}$ in the widely-used Scaled Dot-Product Attention [51] and $i, j$ are the row indices on the matrices $\mathbf{Q}$ and $\mathbf{K}$, respectively. Following [2, 11], we calculate the attention entropy by treating the attention distribution of each token as a probability mass function of a discrete random variable. The attention entropy with respect to the $i^{th}$ row of $\mathbf{A}$ is defined as:

$$\text{Ent}(\mathbf{A}_i) = -\sum_{j=1}^{N} \mathbf{A}_{i,j} \log(\mathbf{A}_{i,j}), \tag{2}$$

where the attention entropy gets its maximum when $\mathbf{A}_{i,j} = 1/N$ for all $j$ and gets its minimum when $\mathbf{A}_{i,j'} = 1$ and $\mathbf{A}_{i,j} = 0$ for any $j \neq j'$. It implies that the attention entropy measures the spatial granularity of information aggregation. A larger attention entropy suggests that wider contextual information is taken into account while a smaller entropy indicates only few tokens are weighed.

We now investigate the relation between the attention entropy and the token number. By substituting Eq.(1) into Eq.(2), we have:

$$\begin{aligned}
\text{Ent}(\mathbf{A}_i) &= -\sum_{j=1}^{N} \Big[ \frac{e^{\lambda \mathbf{Q}_i \mathbf{K}_j^\top}}{\sum_{j'=1}^{N} e^{\lambda \mathbf{Q}_i \mathbf{K}_{j'}^\top}} \log \Big( \frac{e^{\lambda \mathbf{Q}_i \mathbf{K}_j^\top}}{\sum_{j'=1}^{N} e^{\lambda \mathbf{Q}_i \mathbf{K}_{j'}^\top}} \Big) \Big] \\
&\approx \log N + \log \mathbb{E}_j \big( e^{\lambda \mathbf{Q}_i \mathbf{K}_j^\top} \big) - \frac{\mathbb{E}_j \big( \lambda \mathbf{Q}_i \mathbf{K}_j^\top e^{\lambda \mathbf{Q}_i \mathbf{K}_j^\top} \big)}{\mathbb{E}_j \big( e^{\lambda \mathbf{Q}_i \mathbf{K}_j^\top} \big)},
\end{aligned} \tag{3}$$

where $\mathbb{E}_j$ denotes the expectation upon the index $j$, and the last equality holds asymptotically when $N$ gets larger. For the complete proof, please refer to the Supplementary Materials.

Given $\mathbf{X}$ as an encoded sequence from an image, each token $\mathbf{X}_j$ could be assumed as a vector sampled from a multivariate Gaussian distribution $\mathcal{N}(\boldsymbol{\mu}^X, \boldsymbol{\Sigma}^X)$. This prevailing assumption is widely shared in the fields of image synthesis [13] and style transfer [19, 27, 21]. In this way, each $\mathbf{K}_j = \mathbf{X}_j \mathbf{W}^K$ could be considered as a sample from the multivariate Gaussian distribution $\mathcal{N}(\boldsymbol{\mu}^K, \boldsymbol{\Sigma}^K) = \mathcal{N}(\boldsymbol{\mu}^X \mathbf{W}^K, (\mathbf{W}^K)^\top \boldsymbol{\Sigma}^X \mathbf{W}^K)$. In regard to $\mathbf{K}_j$, its dimension-wise linear combination $y_i = \lambda \mathbf{Q}_i \mathbf{K}_j^\top$ could be treated as a scalar random variable sampled from the univariate Gaussian distribution $y_i \sim \mathcal{N}(\mu_i, \sigma_i^2) = \mathcal{N}(\lambda \mathbf{Q}_i (\boldsymbol{\mu}^K)^\top, \lambda^2 \mathbf{Q}_i \boldsymbol{\Sigma}^K \mathbf{Q}_i^\top)$. Under this circumstance, we dive into the expectations $\mathbb{E}_j \big( e^{\lambda \mathbf{Q}_i \mathbf{K}_j^\top} \big)$ and $\mathbb{E}_j \big( \lambda \mathbf{Q}_i \mathbf{K}_j^\top e^{\lambda \mathbf{Q}_i \mathbf{K}_j^\top} \big)$ in Eq.(3):

$$\begin{aligned}
\mathbb{E}_j \big( e^{\lambda \mathbf{Q}_i \mathbf{K}_j^\top} \big) &\doteq \mathbb{E}_j \big( e^{y_i} \big) = e^{\mu_i + \frac{\sigma_i^2}{2}} \\
\mathbb{E}_j \big( \lambda \mathbf{Q}_i \mathbf{K}_j^\top e^{\lambda \mathbf{Q}_i \mathbf{K}_j^\top} \big) &\doteq \mathbb{E}_j \big( y_i e^{y_i} \big) = \big( \mu_i + \sigma_i^2 \big) e^{\mu_i + \frac{\sigma_i^2}{2}},
\end{aligned} \tag{4}$$

For the complete proof, please refer to the Supplementary Materials. By substituting Eq.(4) into Eq.(3), we have:

$$\begin{aligned}
\text{Ent}(\mathbf{A}_i) &= \log N + \log \big( e^{\mu_i + \frac{\sigma_i^2}{2}} \big) - \frac{\big( \mu_i + \sigma_i^2 \big) e^{\mu_i + \frac{\sigma_i^2}{2}}}{e^{\mu_i + \frac{\sigma_i^2}{2}}} \\
&= \log N - \frac{\sigma_i^2}{2}.
\end{aligned} \tag{5}$$

The observation is that the attention entropy varies correspondingly with $N$ in a logarithm magnitude while the latter item $\sigma_i^2 = \lambda^2 \mathbf{Q}_i \boldsymbol{\Sigma}^K \mathbf{Q}_i^\top$ is not relevant with the token number $N$ in the setting of images generation. This implies that each token intends to adaptively deal with wider contextual information for higher resolutions, and narrower contextual information for lower resolutions.

**Interpretations on two observed patterns.** Our key insight lies in the correlation between the fluctuating attention entropy and the emergence of two defective patterns in our observations. Specifically, when the model is synthesizing images with lower resolutions, the attention entropy experiences a statistical decline of $\log N$, leading to a reduced attention on contextual information. Consequently, each token is processing discrete information in isolation and the model struggles to generate objects with intricate details and smooth transitions, giving rise to the first observed pattern.

Conversely, in scenarios involving higher resolutions, the attention entropy exhibits an upward trend, accompanied by the utilization of overwhelming amounts of global information for synthesis. Consequently, each token is processing analogous information and the model generates repetitively disordered presentations, exemplifying the second pattern.

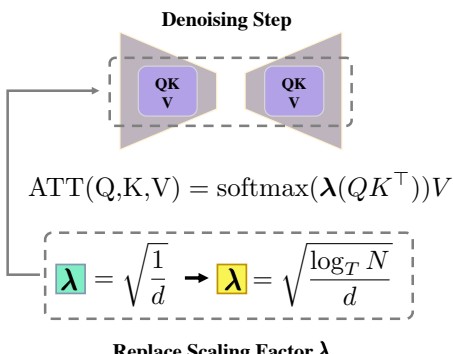

$$\text{ATT(Q,K,V)} = \text{softmax}(\boldsymbol{\lambda}(QK^{\top}))V$$

$$\boldsymbol{\lambda} = \sqrt{\frac{1}{d}} \rightarrow \boldsymbol{\lambda} = \sqrt{\frac{\log_T N}{d}}$$

**Replace Scaling Factor $\boldsymbol{\lambda}$**

Figure 2: The illustration of our proposed process of replacing the scaling factor without additional training, where $T$ and $N$ denote the number of tokens in the training phase and the inference phase.

## 3.2 A scaling factor for mitigating entropy fluctuations

To alleviate the fluctuating entropy with respect to different token numbers $N$ in the inference phase, considering the quadratic form in Eq.(5), we set the scaling hyper-parameter $\lambda$ in the following form:

$$\lambda = c\sqrt{\log N}, \tag{6}$$

where $c$ denotes a constant number. Note that when the token number $N$ is set as the number of tokens during training (denoted as $T$), $\lambda$ should specialize to $1/\sqrt{d}$ for the best stablization of entropy (i.e., $c\sqrt{\log T} \approx 1/\sqrt{d}$). Thus, we have an approximate value of $c$ and substitute it into Eq.(6):

$$\lambda \approx \sqrt{\frac{1}{d\log T}} \cdot \sqrt{\log N} = \sqrt{\frac{\log_T N}{d}}. \tag{7}$$

Intuitively, if the model is dealing with a token number $N > T$, the proposed scaling factor would multiply the feature maps by a factor of squared $\log_T N$ to maintain the rising entropy, and vice versa. Thus, the proposed scaling factor is implemented in a training-free manner, which enjoys better performances in synthesizing images of different resolutions. The illustration of our proposed method is shown in Figure 2.

## 4 Experimental Results

In this section, we evaluate our proposed method on text-to-image synthesis and analyse the efficacy of the proposed scaling factor. The quantitative and qualitative results demonstrate a better performance of our scaling factor. Please refer to the Supplemental Materials for detailed experimental results.

### 4.1 Text-to-image synthesis

**Experimental setup.** We explore the adapting ability of the proposed scaling factor for diffusion models in variable-sized text-to-image synthesis. In particular, we replace the scaling factor of self-attention layers within diffusion models and evaluate their performance without any training upon a subset of `LAION-400M` and `LAION-5B` dataset ([43, 42]), which contain over 400 million and 5.85 billion CLIP-filtered image-text pairs, respectively. We randomly choose text-image pairs out from each dataset and generate images corresponding to texts in multiple settings for evaluation. As for the diffusion models, we consider both Stable Diffusion [37] trained on `LAION-5B` and Latent Diffusion [37] trained on `LAION-400M` to test the performances. The former (which is set to synthesize images with a $512 \times 512$ resolution by default) maintains 4x longer sequences compared with the latter (with a default $256 \times 256$ resolution) across all the layers. For more evaluation results and details, please refer to the Supplementary Materials.

Table 1: **FID scores** ($\downarrow$) for Stable Diffusion and Latent Diffusion in different resolution settings. Stable Diffusion scores 19.8168 for default $512^2$ and Latent Diffusion scores 20.4102 for default $256^2$.

| Stable Diffusion - 512 | | | Latent Diffusion - 256 | | |
|---|---|---|---|---|---|
| Resolution | Original | Ours | Resolution | Original | Ours |
| 224 * 224 | 74.5742 | **41.8925** | 128 * 128 | 65.2988 | **57.5542** |
| 448 * 448 | 19.9039 | **19.4923** | 224 * 224 | 23.0483 | **22.7971** |
| 768 * 768 | 29.5974 | **28.1372** | 384 * 384 | 20.5466 | **20.3842** |
| 512 * 288 | 22.6249 | **21.3877** | 256 * 144 | 33.8559 | **33.0142** |
| 512 * 384 | 20.2315 | **19.8631** | 256 * 192 | 24.2546 | **23.9346** |

Table 2: **CLIP scores** ($\uparrow$) for Stable Diffusion and Latent Diffusion in different resolution settings. Stable Diffusion scores 0.3158 for default $512^2$ and Latent Diffusion scores 0.3153 for default $256^2$.

| Stable Diffusion - 512 | | | Latent Diffusion - 256 | | |
|---|---|---|---|---|---|
| Resolution | Original | Ours | Resolution | Original | Ours |
| 224 * 224 | 0.2553 | **0.2764** | 128 * 128 | 0.2679 | **0.2747** |
| 448 * 448 | 0.3176 | **0.3180** | 224 * 224 | 0.3094 | **0.3096** |
| 768 * 768 | 0.3142 | **0.3152** | 384 * 384 | 0.3148 | **0.3156** |
| 512 * 288 | 0.3047 | **0.3066** | 256 * 144 | 0.2884 | **0.2890** |
| 512 * 384 | 0.3134 | **0.3139** | 256 * 192 | 0.3070 | **0.3075** |

Table 3: **Results of human evaluation.** We report the score of consistency ($\uparrow$) for the text-to-image results of Stable Diffusion in different resolution settings.

| Resolution | 224 * 224 | 448 * 448 | 768 * 768 | 512 * 288 | 512 * 384 |
|---|---|---|---|---|---|
| Original | 5.13 | 6.45 | 6.74 | 6.11 | 6.75 |
| Ours | **6.77** | **7.12** | **7.56** | **7.42** | **6.98** |

**Quantitative comparison.** We compare the performance of the proposed scaling factor against the scaling factor $\lambda = 1/\sqrt{d}$ for Stable Diffusion [37] and Latent Diffusion [37] in multiple squared resolution settings (small, medium and large sizes according to their default training sizes). We evaluate them with the metric of Fréchet Inception Distance (FID) [13], which measures image quality and diversity. As shown in Table 1, our scaling factor universally improves FID for both Stable Diffusion and Latent Diffusion in all resolution settings, especially for the images with small resolution (a significant improvement from 74.6 to 41.9 for Stable Diffusion on the resolution $224 \times 224$). It is worth noting that the proposed method needs no additional training and achieves these in a plug-and-play manner with trivial complexity.

Besides FID, we also evaluate the degree of semantic alignment between text prompts and the corresponding synthesized images with CLIP scores [35], which utilizes contrastive learning to establish connections between visual concepts and natural language concepts. As shown in Table 2, our scaling factor achieves better CLIP scores than the original one in all resolution settings. It shows that our scaling factor performs better in the alignment of text prompts and the synthesized images.

Additionally, we present the results of user study in Table 3. We conduct a text-based pairwise preference test. That is, for a given sentence, we pair the image from our model with the synthesized image from the baseline model and ask 45 annotators in total to give each synthesized image a rating score for consistency. For each human annotators, we pay $15 for effective completeness of user study evaluation. We observe that users rate higher for our method, which suggests that the synthesized images from our method are more contextually coherent with texts than the baseline. Besides, with the refinement from the proposed scaling factor, the generated contents from our model are able to convey more natural and informative objects.

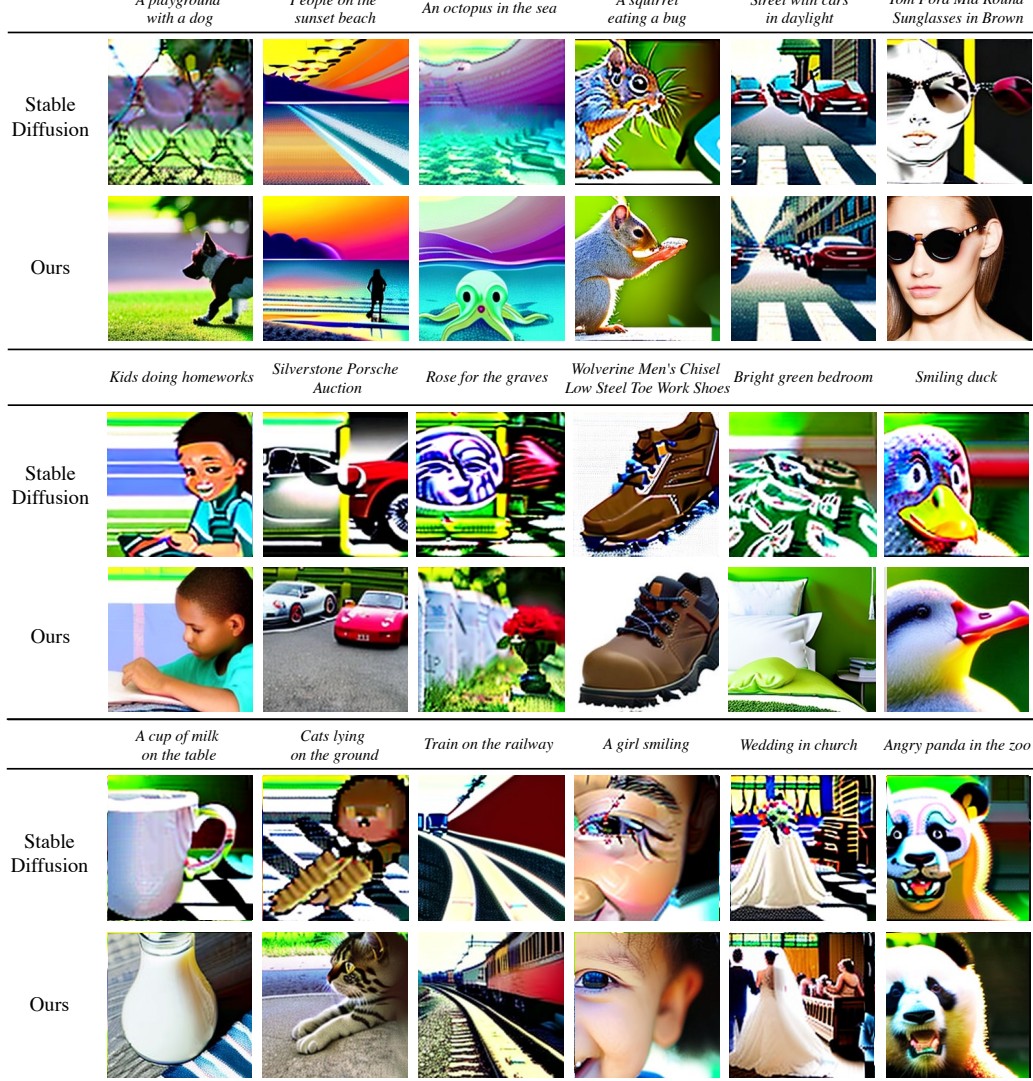

Figure 3: **Qualitative comparison** on the scale factors for the resolution $224 \times 224$. The original scaling factor misses out content or roughly depicts the objects in prompts while our scaling factor manages to synthesize visual concepts in high fidelity and better illumination. Please zoom in for better visual effect.

**Qualitative comparison.** In Figures 3 and 4, we show some visualization results for the qualitative comparison between two scaling factors with Stable Diffusion. Paired images are synthesized with the same seeds and text prompts. Please zoom in for the best visual effect. For Figure 3, we demonstrate results under the proposed and the original scaling factors for Stable Diffusion with $224 \times 224$ resolution, which is lower than the default training resolution $512 \times 512$ of Stable Diffusion. We observe that in the original Stable Diffusion setting, objects are either missed out (e.g., absent dog, people and octopus) or roughly depicted (e.g., crudely rendered squirrel, cars and person with sunglasses). Our interpretation is that for lower resolutions, according to Eq. (5), the model is synthesizing images with a less entropy during the inference phase when compared to its training phase, which implies that the model is suffering from deprived information. In this way, the model has more difficulty in generating complex objects and cannot generate attractive images. As for the results generated by the proposed scaling factor, we show that images are synthesized with better alignments with prompts (e.g., clear presence of dog, people and octopus) and objects are generated more vividly (e.g., realistic squirrel, cars and person with sunglasses). Note that our scaling factor also performs better in the illumination and shading because of the utilization of more aggregated

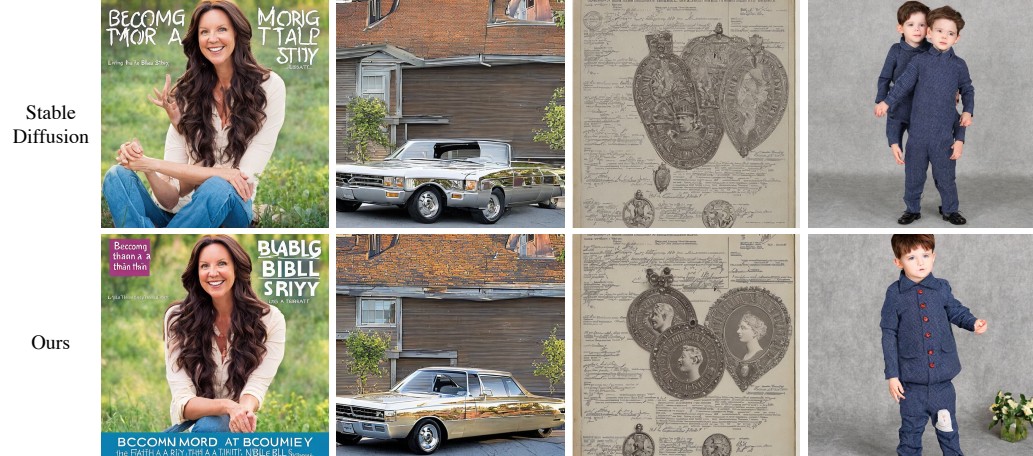

Figure 4: **Qualitative comparison** on the scale factors for the resolution $768 \times 768$. The original scaling factor generates objects in repeated and messy patterns while our scaling factor manages to spatially arrange the visual concepts in clear order. Please zoom in for better visual effect.

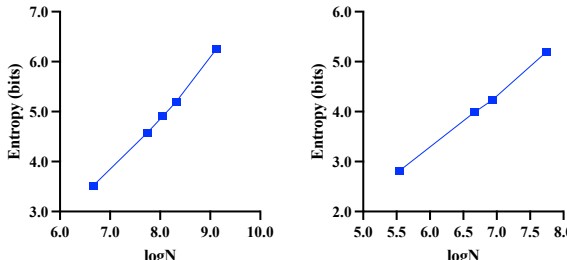

Figure 5: **Analysis results** on the average attention entropy with respect to the number of tokens for Stable Diffusion model (left) and latent diffusion model (right). $N$ refers to the number of tokens. As shown in the figure, average attention entropy has a linear relationship with the logarithm of $N$, which confirms the effectiveness of our asymptotic equality in Eq.(5)

and contextualized information with higher entropy. Considering its flexibility, our method might offer valuable insights into the fields of inpainting [28, 56] and compositional scene generation [59].

For Figure 4, we demonstrate the results of Stable Diffusion under the proposed and the original scaling factors with the $768 \times 768$ resolution, which is higher than the default training resolution $512 \times 512$ of Stable Diffusion. Different from the rough objects in Figure 3, we note that objects are depicted with overwhelming details in a messy and disordered manner (e.g., repeated hands, a extended vehicle and messy seals). The reason is that for higher resolutions, according to Eq.(5), the model are bearing more entropy than it does in the training period. As a result, tokens are attending to numerous but redundant global information and the model depicts objects with burdensome details, leading to the patterns observed. For the results by our scaling factor, we note that the repeated patterns are greatly alleviated (e.g., a pair of normal hands and a regular vehicle) and synthesized objects are spatially arranged in order without losing the details (three seals well depicted in order).

## 4.2 Analysis

In this section, we conduct a comprehensive analysis of the practical implications of our theoretical findings. This analysis is of great importance due to the nature of our fundamental theoretical discovery outlined in Eq.(5) . As an asymptotic equality, it might exhibit limitations in practical scenarios where the token quantity, denoted as $N$, falls below a certain threshold. Thus, it becomes imperative to assess the feasibility and the applicability of our theoretical finding under such circumstances. Subsequently, we delve into an in-depth examination of the pivotal role played by the scaling factor

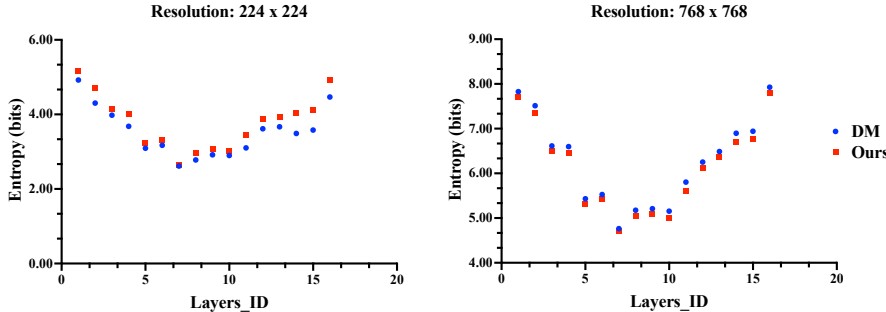

Figure 6: **Analysis results** on the average attention entropy with respect to two scale factors for two resolution. Layer ID refers to the index number of self-attention layers in the model and DM stands for original diffusion model. The figure illustrate that the proposed scaling factor mitigates the fluctuation of attention entropy.

in mitigating the fluctuations within the attention entropy, as presented in Section 3.2. This rigorous analysis serves to substantiate and validate the insights elucidated in the aforementioned section. To support our claims, we present visually intuitive representations of the empirical results, which unequivocally demonstrate the alignment between our statements and the experimental outcomes.

**The relationship between the attention entropy and the token number.**    We conduct an experiment on text-to-image tasks to investigate the relation between the entropy and the token number $N$, which is statistically connected in Eq.(5). In particular, we synthesize images with various resolutions (including $224^2, 384^2, 448^2, 512^2$ and $768^2$ for Stable Diffusion and $128^2, 224^2, 256^2$ and $384^2$ for Latent Diffusion model) and calculate their attention entropy during the inference period. We average the results across attention heads and layers for 5K samples on each resolution. Figure 5 illustrates that the average attention entropy has a linear relationship with the logarithm of the quantity of tokens for both diffusion models, which accords with their asymptotic relationship proposed in Eq.(5). The visualized results provide the empirical evidence that substantiates the efficacy of our theoretical findings and the feasible extensions for those resolutions close to the mentioned resolutions.

**The role of scaling factor for stabilizing the attention entropy.**    We conduct an experiment on the mitigating effect of the proposed scaling factor onto the fluctuated attention entropy. Specifically, with two scaling factors, we synthesize images with a lower and a higher resolution (including $224^2$ and $768^2$ for Stable Diffusion) and record their attention entropy for each attention layers. We average the results across attention heads for 5K samples. As demonstrated in the Figure 6, the proposed scaling factor does contribute to an increase in the synthesis process for the resolution $224^2$ and a decrease in that for the resolution $768^2$, which accords with our motivation in Section 3.2 and shows the efficacy of our scaling factor in mitigating the fluctuation of the attention entropy.

### 4.3   Difference with other candidate methods

In this section, we discuss upon the difference between the proposed method and other candidate methods for variable-sized image synthesis, including up/down-sampling methods and super-resolution methods. While all the mentioned methods share the same goal to synthesis images of variable sizes, there are three distinctive features setting our method apart from the rest.

**Different aspect ratio.**    Other candidate methods do not support diffusion models to generate new images with a different aspect ratio. In comparison, our method could improve image synthesis in different aspect ratio, validated by both qualitative and quantitative experiments in Section 4.1.

**Richness of visual information (important for higher resolutions).**    When users generate images with higher resolutions, what they are expecting is not only more pixels but also richer semantic information in images. Under this circumstance, our method augments semantic information by enabling models to deal with more tokens in an adaptive manner, which enables images to possess a more extensive range of content. As shown in Figure 7, our method generates images with more

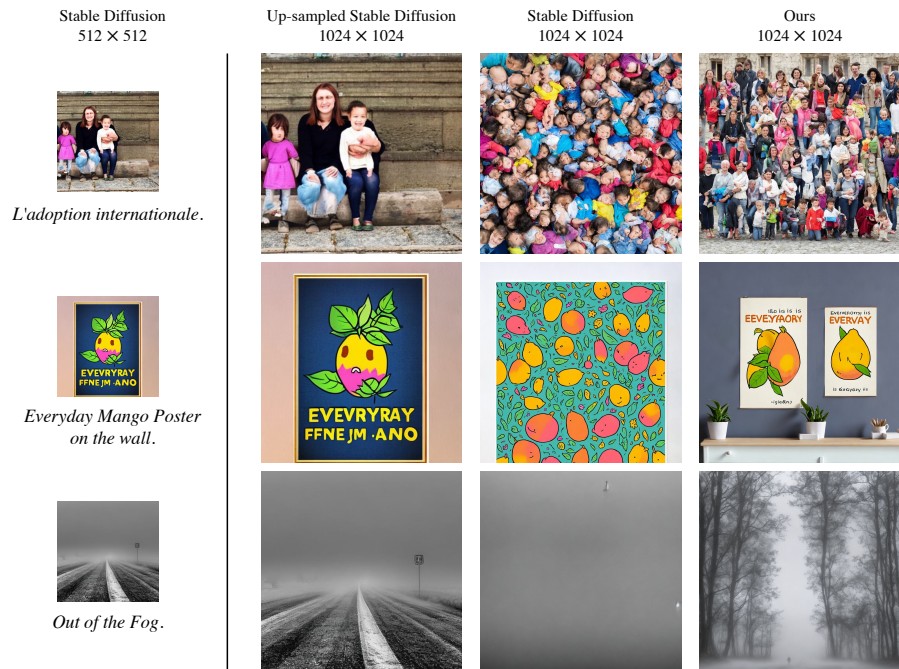

| Stable Diffusion
$512 \times 512$ | Up-sampled Stable Diffusion
$1024 \times 1024$ | Stable Diffusion
$1024 \times 1024$ | Ours
$1024 \times 1024$ |

*L'adoption internationale.*

*Everyday Mango Poster on the wall.*

*Out of the Fog.*

Figure 7: **Qualitative results** to illustrate the difference in visual information richness between up-sampled methods, the original Stable Diffusion and the proposed method. Note that our method (right) introduces more visual information with a different level of granularity compared with up-sampling and super-resolution methods (left). Additionally, our method could better deal with repetitively disordered pattern emerging in the original Stable Diffusion model (middle).

richness and expressiveness in semantics. In contrast, super-resolution and other methods scale up the original images and focus on better image clarity and finer details instead of richer semantics, introducing visual information at a different granularity level compared with the proposed method.

**Time cost and memory usage (important for lower resolutions).** For diffusion models adapted by our method, their time cost and spatial usage become proportional to the generation resolution, while down-sampling methods are constantly constrained to the fixed cost brought by training resolutions. As a result, our method could efficiently enable low resolution image synthesis especially on portable devices, which has a high demand for both time cost and memory usage other than image fidelity. For more quantitative results to support this, please refer to the Supplementary Materials.

## 5   Conclusion

In this work, we delves into a new perspective of adapting diffusion models to effectively synthesize images of varying sizes with superior visual quality, based on the concept of entropy. We establish a statistical connection between the number of tokens and the entropy of attention maps. Utilizing this connection, we give interpretations to the defective patterns we observe and propose a novel scaling factor for visual attention layers to better handle sequences of varying lengths. The experimental results demonstrate that our scaling factor effectively enhances both quantitative and qualitative scores for text-to-image of varying sizes in a training-free manner, while stabilizing the entropy of attention maps for variable token numbers. Moreover, we provide extensive analysis and discussion for the efficiency and the significance of our method.

**Limitations**   One limitation of this study pertains to the metrics used for quantitative evaluation. While these metrics primarily evaluate the quality of the generated content, there is a lack of methodology for evaluating the fidelity of images for different resolutions. The same issue applies for the evaluation of repetitively disordered pattern, which is in part compensated by qualitative scores.

## Acknowledgment

This work was supported in part by the National Key R&D Program of China (No.2021ZD0112803), the National Natural Science Foundation of China (No.62176060, No.62176061), STCSM project (No.22511105000), Shanghai Municipal Science and Technology Major Project (2021SHZDZX0103), UniDT's Cognitive Computing and Few Shot Learning Project, and the Program for Professor of Special Appointment (Eastern Scholar) at Shanghai Institutions of Higher Learning.

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
