# Training-free Diffusion Model Adaptation for Variable-Sized Text-to-Image Synthesis (Supplementary Materials)

**Zhiyu Jin**[1], **Xuli Shen**[1,2], **Bin Li**[1]*, **Xiangyang Xue**[1]
[1] Shanghai Key Laboratory of Intelligent Information Processing
School of Computer Science, Fudan University
[2] UniDT Technology

In the Supplementary Materials, we first provide the proofs of the main conclusions in Section 3, then we introduce the involved existing assets, evaluation settings and computing infrastructure, and finally we show the additional experimental results on our method for the performance comparison.

## Contents

## 1 Proofs

In this section, we provide the proofs of the connection between attention entropy and token number, which constitutes the main conclusion in Section 3.

Let $\mathbf{X} \in \mathbb{R}^{N \times d}$ denotes an image token sequence to an attention module, where $N, d$ are the number of tokens and the token dimension, respectively. We then denote the key, query and value matrices as $\mathbf{K} = \mathbf{X}\mathbf{W}^K, \mathbf{Q}$ and $\mathbf{V}$, where $\mathbf{W}^K \in \mathbb{R}^{d \times d_r}$ is a learnable projection matrix. The attention layer ([2, 11]) computes $\text{Attention}(\mathbf{Q}, \mathbf{K}, \mathbf{V}) = \mathbf{A}\mathbf{V}$ with the attention map $\mathbf{A}$ calculated by the row-wise softmax function as follows:

$$\mathbf{A}_{i,j} = \frac{e^{\lambda \mathbf{Q}_i \mathbf{K}_j^\top}}{\sum_{j'=1}^{N} e^{\lambda \mathbf{Q}_i \mathbf{K}_{j'}^\top}}, \tag{1}$$

---

*Corresponding author

37th Conference on Neural Information Processing Systems (NeurIPS 2023).

where $\lambda$ is a scaling factor, usually set as $1/\sqrt{d}$ in the widely-used Scaled Dot-Product Attention [11] and $i, j$ are the row indices on the matrices $\mathbf{Q}$ and $\mathbf{K}$, respectively. Following [1, 4], we calculate the attention entropy by treating the attention distribution of each token as a probability mass function of a discrete random variable. The attention entropy with respect to $i^{th}$ row of $\mathbf{A}$ is defined as:

$$\text{Ent}(\mathbf{A}_i) = -\sum_{j=1}^{N} \mathbf{A}_{i,j} \log(\mathbf{A}_{i,j}), \tag{2}$$

We now investigate the relation between the attention entropy and the token number. By substituting Eq.(1) into Eq.(2), we have:

$$
\begin{aligned}
\text{Ent}(\mathbf{A}_i) &= -\sum_{j=1}^{N} \Big[ \frac{e^{\lambda \mathbf{Q}_i \mathbf{K}_j^\top}}{\sum_{j'=1}^{N} e^{\lambda \mathbf{Q}_i \mathbf{K}_{j'}^\top}} \log\Big( \frac{e^{\lambda \mathbf{Q}_i \mathbf{K}_j^\top}}{\sum_{j'=1}^{N} e^{\lambda \mathbf{Q}_i \mathbf{K}_{j'}^\top}} \Big) \Big] \\
&= \log \sum_{j'=1}^{N} e^{\lambda \mathbf{Q}_i \mathbf{K}_{j'}^\top} - \frac{\sum_{j=1}^{N} \big( e^{\lambda \mathbf{Q}_i \mathbf{K}_j^\top} \lambda \mathbf{Q}_i \mathbf{K}_j^\top \big)}{\sum_{j'} e^{\lambda \mathbf{Q}_i \mathbf{K}_{j'}^\top}} \\
&= \log N + \log \big( \frac{1}{N} \sum_{j=1}^{N} e^{\lambda \mathbf{Q}_i \mathbf{K}_j^\top} \big) - \frac{\frac{1}{N} \sum_{j=1}^{N} \big( e^{\lambda \mathbf{Q}_i \mathbf{K}_j^\top} \lambda \mathbf{Q}_i \mathbf{K}_j^\top \big)}{\frac{1}{N} \sum_{j} e^{\lambda \mathbf{Q}_i \mathbf{K}_j^\top}} \\
&\approx \log N + \log \mathbb{E}_j \big( e^{\lambda \mathbf{Q}_i \mathbf{K}_j^\top} \big) - \frac{\mathbb{E}_j \big( \lambda \mathbf{Q}_i \mathbf{K}_j^\top e^{\lambda \mathbf{Q}_i \mathbf{K}_j^\top} \big)}{\mathbb{E}_j \big( e^{\lambda \mathbf{Q}_i \mathbf{K}_j^\top} \big)},
\end{aligned}
\tag{3}
$$

where $\mathbb{E}_j$ denotes the expectation upon the index $j$, and the last equality holds asymptotically when $N$ gets larger.

Given $\mathbf{X}$ as an encoded sequence from an image, we assume each token $\mathbf{X}_j$ as a vector sampled from a multivariate Gaussian distribution $\mathcal{N}(\boldsymbol{\mu}^X, \boldsymbol{\Sigma}^X)$. This widespread assumption is shared by multiple works [5, 6, 7] in image synthesis and style transfer. In this way, each $\mathbf{K}_j = \mathbf{X}_j \mathbf{W}^K$ could be considered as a sample from a multivariate Gaussian distribution $\mathcal{N}(\boldsymbol{\mu}^K, \boldsymbol{\Sigma}^K) = \mathcal{N}(\boldsymbol{\mu}^X \mathbf{W}^K, (\mathbf{W}^K)^\top \boldsymbol{\Sigma}^X \mathbf{W}^K)$. In regard to $\mathbf{K}_j$, its dimension-wise linear combination $y_i = \lambda \mathbf{Q}_i \mathbf{K}_j^\top$ could be treated as a scalar random variable sampled from a Gaussian distribution $y_i \sim \mathcal{N}(\mu_i, \sigma_i^2) = \mathcal{N}(\lambda \mathbf{Q}_i (\boldsymbol{\mu}^K)^\top, \lambda^2 \mathbf{Q}_i \boldsymbol{\Sigma}^K \mathbf{Q}_i^\top)$. Under this circumstance, we dive into the expectations $\mathbb{E}_j \big( e^{\lambda \mathbf{Q}_i \mathbf{K}_j^\top} \big)$ and $\mathbb{E}_j \big( \lambda \mathbf{Q}_i \mathbf{K}_j^\top e^{\lambda \mathbf{Q}_i \mathbf{K}_j^\top} \big)$ in Eq.(3):

$$
\begin{aligned}
\mathbb{E}_j \big( e^{\lambda \mathbf{Q}_i \mathbf{K}_j^\top} \big) &\doteq \mathbb{E}_j \big( e^{y_i} \big) \\
&= \int e^{y_i} \frac{1}{\sqrt{2\pi}\sigma_i} e^{-\frac{y_i^2 - 2\mu_i y_i + \mu_i^2}{2\sigma_i^2}} dy_i \\
&= \int \frac{1}{\sqrt{2\pi}\sigma_i} e^{-\frac{(y_i - \mu_i - \sigma_i^2)^2}{2\sigma_i^2}} e^{\mu_i + \frac{\sigma_i^2}{2}} dy_i \\
&= e^{\mu_i + \frac{\sigma_i^2}{2}} \\
\mathbb{E}_j \big( \lambda \mathbf{Q}_i \mathbf{K}_j^\top e^{\lambda \mathbf{Q}_i \mathbf{K}_j^\top} \big) &\doteq \mathbb{E}_j \big( y_i e^{y_i} \big) \\
&= \int y_i e^{y_i} \frac{1}{\sqrt{2\pi}\sigma_i} e^{-\frac{y_i^2 - 2\mu_i y_i + \mu^2}{2\sigma^2}} dy_i \\
&= \int y_i \frac{1}{\sqrt{2\pi}\sigma_i} e^{-\frac{(y_i - \mu_i - \sigma_i^2)^2}{2\sigma_i^2}} e^{\mu_i + \frac{\sigma_i^2}{2}} dy_i \\
&= \big( \mu_i + \sigma_i^2 \big) e^{\mu_i + \frac{\sigma_i^2}{2}},
\end{aligned}
\tag{4}
$$

By substituting Eq.(4) into Eq.(3), we have:

$$
\begin{aligned}
\mathrm{Ent}(\mathbf{A}_i) &= \log N + \log\left(e^{\mu_i + \frac{\sigma_i^2}{2}}\right) - \frac{\left(\mu_i + \sigma_i^2\right)e^{\mu_i + \frac{\sigma_i^2}{2}}}{e^{\mu_i + \frac{\sigma_i^2}{2}}} \\
&= \log N + \mu_i + \frac{\sigma_i^2}{2} - (\mu_i + \sigma_i^2) \\
&= \log N - \frac{\sigma_i^2}{2}.
\end{aligned}
\tag{5}
$$

In this way, we reach the connection between attention entropy and token number, which lays a solid theoretical foundation for the subsequent interpretations and the scaling factor proposed.

## 2 Implementation details

In this section, we provide the detailed implementation settings in Section 4, including involved existing assets, evaluation settings, computing infrastructure, etc.

### 2.1 Involved existing assets.

For the evaluation code, we refer to the github code repository called Diffusers (URL) released by Hugging face with Apache License, Version 2.0. The revised code are shown in Algorithm 1.

For the model parameters, we select the stable diffusion parameters released by stability AI ([8], version: stable-diffusion-2-1-base, URL) and the latent diffusion parameters released by CompVis ([8], version: ldm-text2im-large-256, URL). Both of them are top-ranked parameter files for downloading.

For datasets, we use `LAION-400M` ([10], URL) and `LAION-5B` dataset ([9], URL), which contain over 400 million and 5.85 billion CLIP-filtered image-text pairs, respectively.

---

**Algorithm 1** Replace the scaling factor

---

**Require:** Token number during training $\mathbf{T} \geq 0,$ Flag to replace the scaling factor $C$
**Ensure:** $\lambda = \sqrt{\log_T N/d}$
   $\mathbf{Q} \leftarrow \mathbf{XW}_Q$
   $\mathbf{K} \leftarrow \mathbf{XW}_K$
   $\mathbf{V} \leftarrow \mathbf{XV}_V$
   **if** $C$ is `True` **then**
      $\lambda \leftarrow \sqrt{\log_T N/d}$
   **else if** $C$ is `False` **then**
      $\lambda \leftarrow \sqrt{1/d}$
   **end if**
   $\mathrm{Attn} = \mathtt{softmax}(\lambda \mathbf{Q}\mathbf{K}^\top)\mathbf{V}$

---

### 2.2 Evaluation settings

We evaluate two scaling factors on a subset of `LAION-400M` and `LAION-5B` dataset. We randomly select 30K image-text data pairs from `LAION-5B` and 50K image-text data pairs from `LAION-400M`, respectively. We then synthesize corresponding 30K images upon `LAION-5B` text data with stable diffusion model and 50K images upon `LAION-400M` text data with latent diffusion model. We compute corresponding FID-30K (Stable diffusion) and FID-50k (Latent diffusion) with synthesized images and original images from datasets.

### 2.3 Computing infrastructure

Experiments are conducted on a server with Intel(R) Xeon(R) Gold 6226R CPUs @ 2.90GHz and four NVIDIA Tesla A100 GPUs. The code is developed based on the PyTorch framework with version 1.9.1.

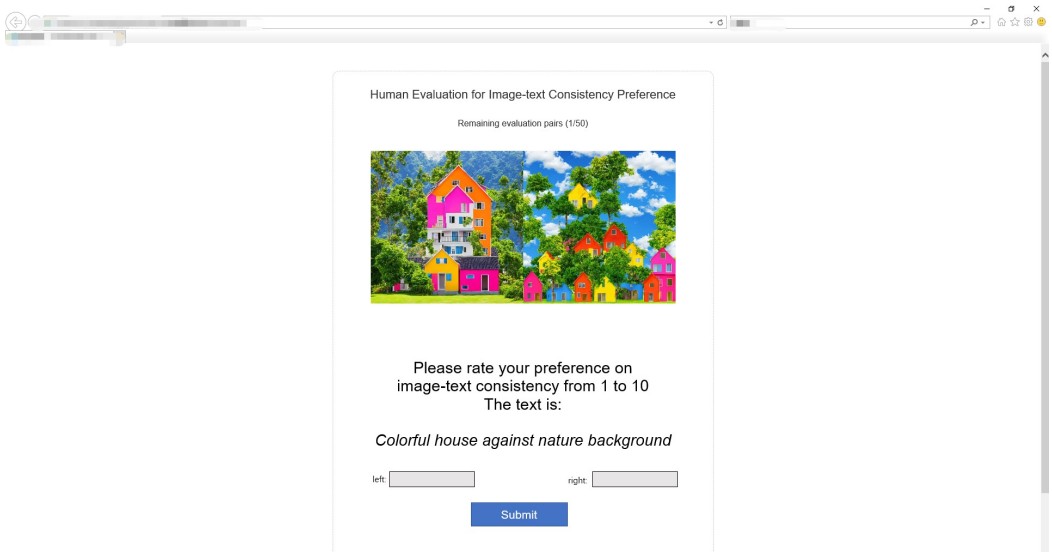

Figure 1: The screenshot of our human evaluation website.

## 2.4 Human evaluation details

We conduct an text-based pairwise preference test. For a specific context, we pair the synthesized image of our method with the synthesized image from the baseline and ask 45 annotators to give each response a rating score for consistency. We randomly select 50 pairs of images-text data (two images from two methods and a corresponding text prompt) and show them to 45 participants to rate from 1 to 10. We observe that users rate higher for our method, which suggests that the synthesized images from our method are more contextually coherent with texts than the baseline. Besides, with the refinement from the proposed scaling factor, the generated contents from our model are able to convey more natural and informative objects. The screenshot is depicted in Figure 1.

## 3 Broader impacts

Our approach focuses on synthesizing images with a high level of fidelity, which raises concerns about potential impacts in a broader context, specifically related to the misuse of portraits. Notably, when our method is provided with prompts referencing highly popular celebrities, it consistently generates figures that bear a strong resemblance to the mentioned individuals. Consequently, there is a risk of portrait misusing or even the infringement of portrait rights.

It is worth acknowledging that the potential impact mentioned above is not unique to our method but is a challenge faced by most image synthesis techniques. We firmly believe that numerous researchers have already recognized and been working on this issue with utmost seriousness and diligence.

## 4 Additional experimental results

In this section, we provide more qualitative and quantitative results for the validation of our theoretical findings and the proposed scaling factor.

### 4.1 Qualitative results

As shown in Figure 2, 3 and 4, our scaling factor manages to synthesize objects with better visual effect in lower resolution images, outperforming the original scaling which depicts visual concepts in a rough manner. From Figure 5 and 6, our scaling factor does better in naturally organizing visual concepts portrayal in higher resolution images, surpassing the original scaling factor in image presentations.

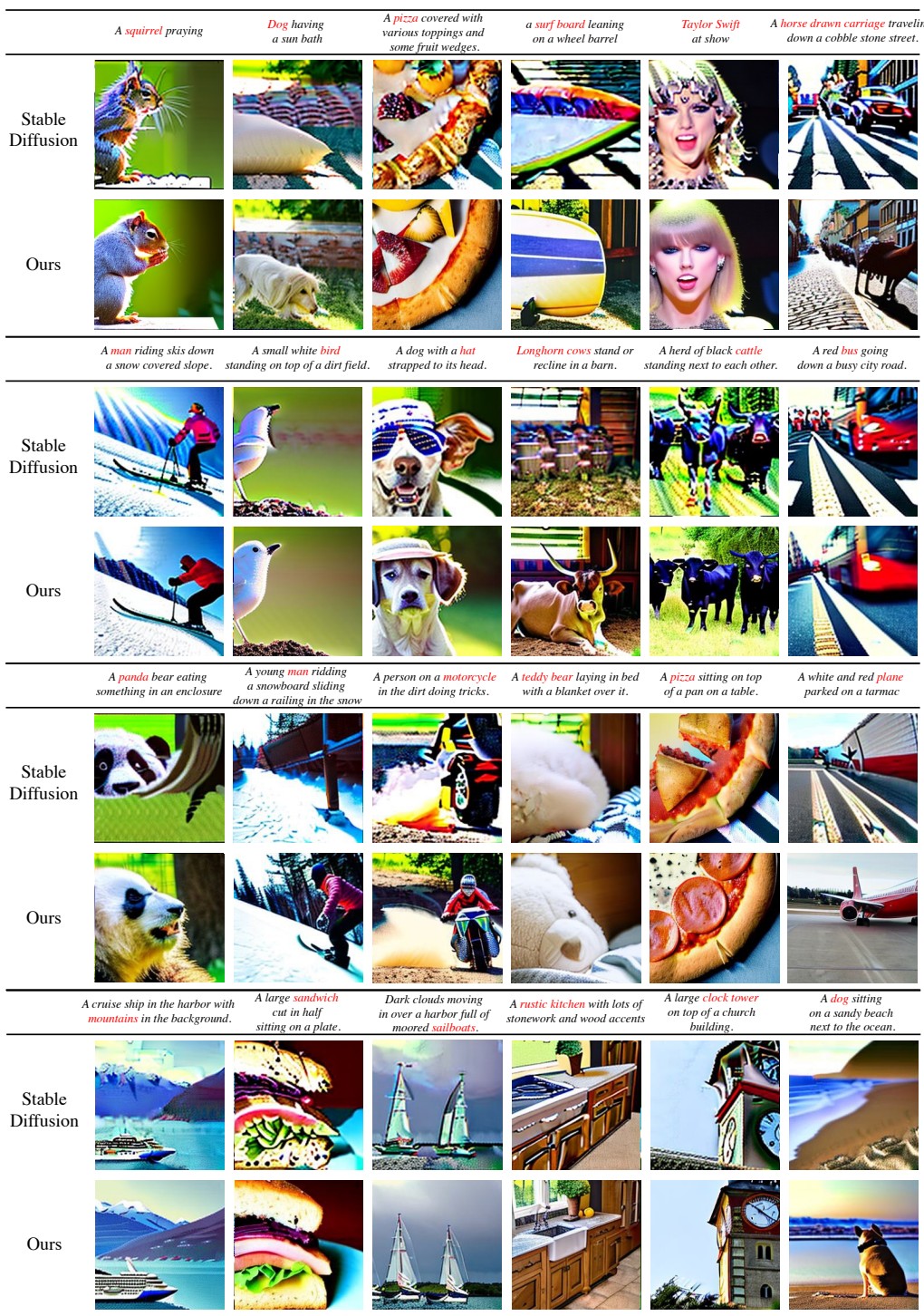

Figure 2: **Qualitative comparison** on the scale factors for the resolution $224 \times 224$. The original scaling factor misses out content or roughly depicts the objects in prompts while our scaling factor manages to synthesize visual concepts in high fidelity and better illumination. Please zoom in for better visual effect.

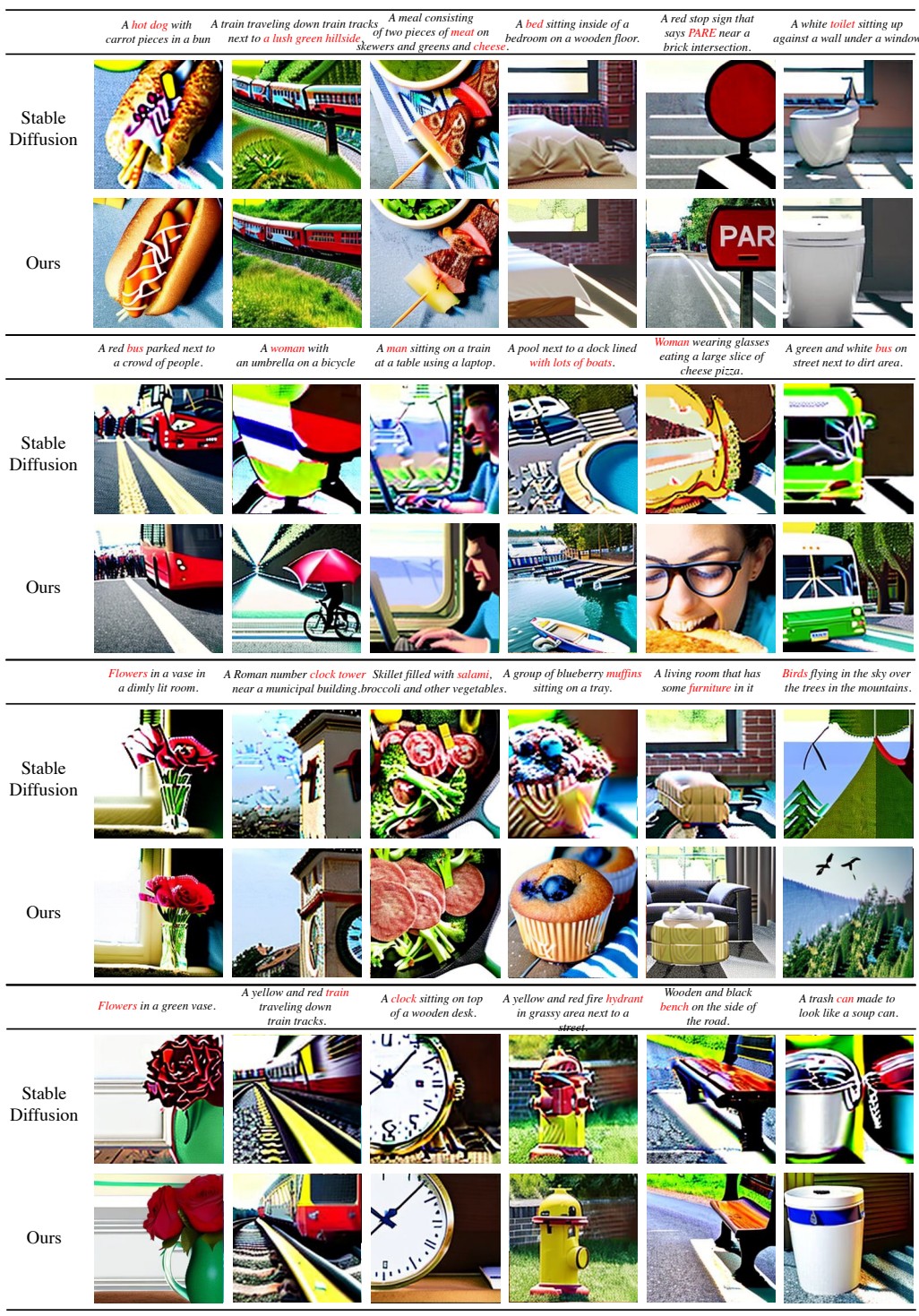

Figure 3: **Qualitative comparison** on the scale factors for the resolution $224 \times 224$. The original scaling factor misses out content or roughly depicts the objects in prompts while our scaling factor manages to synthesize visual concepts in high fidelity and better illumination. Please zoom in for better visual effect.

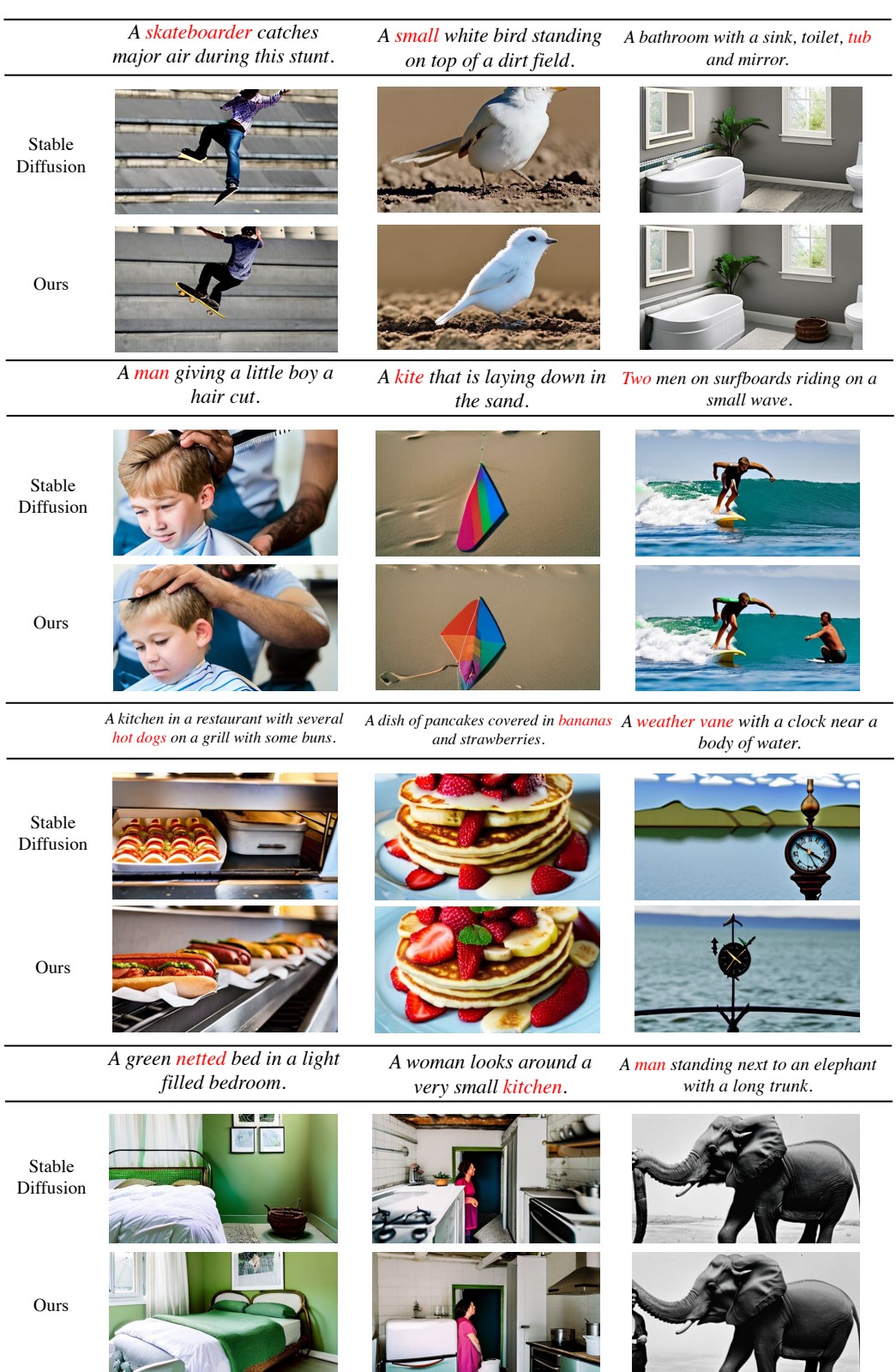

Figure 4: **Qualitative comparison** on the scale factors for the resolution $512 \times 288$. The original scaling factor misses out content or roughly depicts the objects in prompts while our scaling factor manages to synthesize visual concepts in high fidelity and better illumination. Please zoom in for better visual effect.

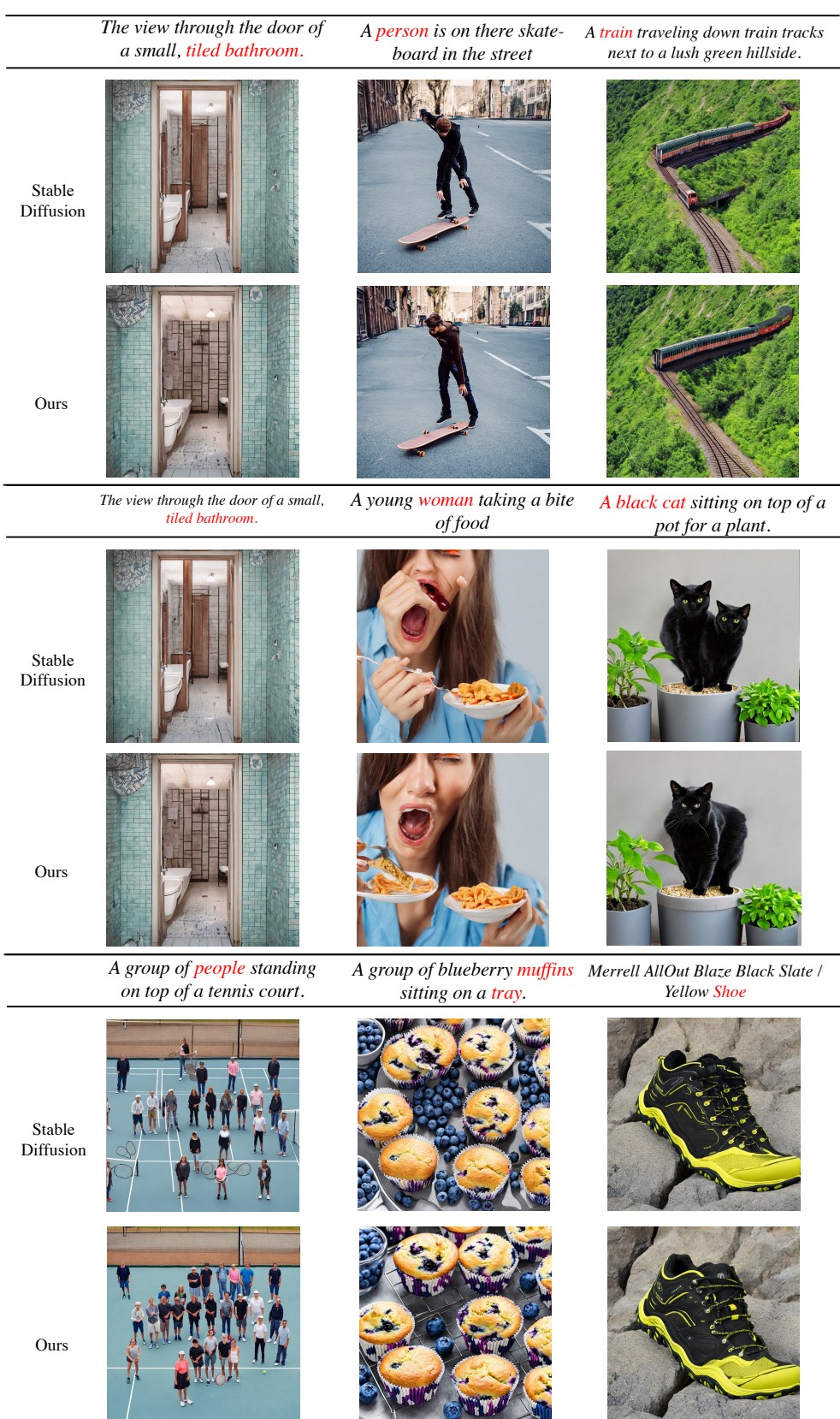

Figure 5: **Qualitative comparison** on the scale factors for the resolution $768 \times 768$. The original scaling factor depicts the objects in a repetitive and unorganized pattern while our scaling factor manages to synthesize visual concepts in high fidelity. Please zoom in for better visual effect.

Stable
Diffusion

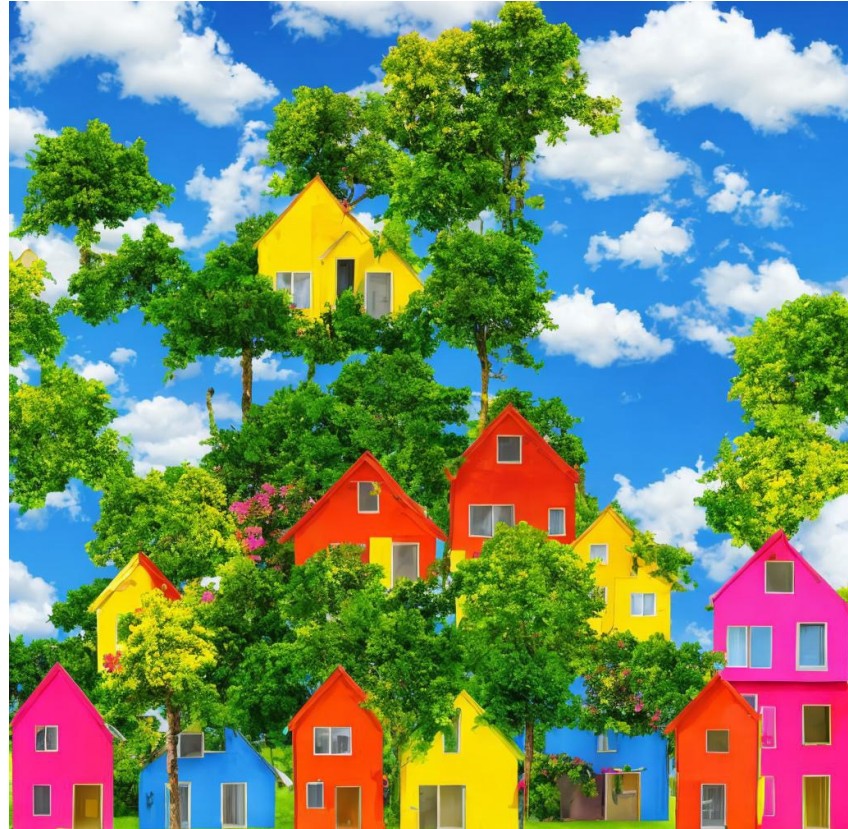

*Prompt: Colorful house against nature background.*

Ours

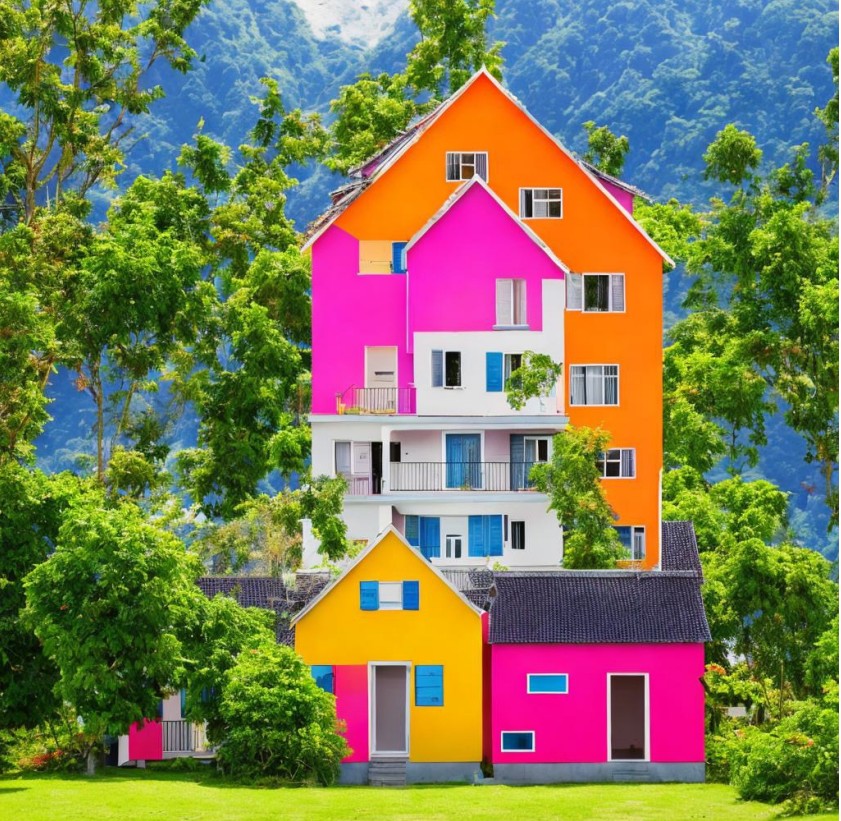

Figure 6: **Qualitative comparison** on the scale factors for the resolution $1024 \times 1024$.

Table 1: FID, memory usage and time cost of stable diffusion with multiple settings for different resolutions. The results for low resolution demonstrate that our method could achieve better performance in memory usage and time cost with trade-off in fidelity.

| Resoluions | FID ($\downarrow$) | GPU Memory (MiB) ($\downarrow$) | Average Time (s) ($\downarrow$) |
|---|---|---|---|
| 128 * 128 (Original) | 191.5447 | **6867** | **3.4953** |
| 128 * 128 (Ours) | 127.2798 | **6867** | 3.4955 |
| 512 * 512 and Resize to 128 | **36.0742** | 9324 | 17.4377 |
| 224 * 224 (Original) | 74.5742 | **7064** | **5.6268** |
| 224 * 224 (Ours) | 41.8925 | **7064** | 5.6274 |
| 512 * 512 and Resize to 224 | **21.8415** | 9324 | 17.4457 |

Table 2: FID (4K samples), memory usage and time cost of our method and MultiDiffusion[3]. Our method outperforms MultiDiffusion on each metric except for the GPU Memory.

| Resoluions | FID ($\downarrow$) | GPU Memory (MiB) ($\downarrow$) | Average Time (s) ($\downarrow$) |
|---|---|---|---|
| 224 * 224 (Ours) | **50.3515** | 7064 | **5.6274** |
| 224 * 224 (MD) | 154.6925 | **7061** | 31.7680 |
| 768 * 768 (Ours) | **28.1372** | 19797 | **36.7140** |
| 768 * 768 (MD) | 40.9270 | **9906** | 122.2485 |

## 4.2 Quantitative results

In Table 1, we present FID, memory usage and time cost of stable diffusion with multiple settings for the text-to-image synthesis of various resolutions ($128^2, 224^2, 768^2, 1024^2$). For low resolution, our method achieves better performance in memory usage and time cost with trade-off in fidelity. Considering the high demand for memory and time resources (especially for portable devices), we believe the trade-off is acceptable.

In Table 2, We compare our method with MultiDiffusion[3] and report corresponding FID(4K samples), memory usage and time cost. As it is shown, our method outperforms MultiDiffusion on each metric except for the GPU Memory.