# OpenReview forum: "Training-free Diffusion Model Adaptation for Variable-Sized Text-to-Image Synthesis"
_NeurIPS.cc/2023/Conference — NeurIPS 2023 poster_

### Official Review · Reviewer_XLhR · 2023-07-06

**Soundness:** 3 good
**Presentation:** 4 excellent
**Contribution:** 3 good
**Rating:** 7
**Confidence:** 4

**Summary:**

This paper proposes both an analysis and a contribution to fix a problem found during the analysis.

They start with the premise that diffusion models should be able to generate arbitrary size images, and training specialized models for each image size is too expensive, which is correct. Using diffusion models trained for square images would be much cheaper and easier.
They identify two key problems when using square models to generate arbitrary aspect ratio images: incomplete/inadequate objects and repetitive/disorganized patterns.
In order to improve performance all around (quality, prompt following, etc.) they study entropy in the generated image.
Specifically, they note that as entropy rises tokens attend to wider regions, and relate this phenomenon to the problems delineated above.
Finally, they find that simply proposing a scaling factor mitigates many of these issues.

**Strengths:**

1. The idea to add a scaling factor that counteracts entropy fluctuation in a training-free manner is really nice and the formulation is a principled approximation that is easily computable.
2. The implementation is incredibly simple, a large plus.
3. The work presents strong quantitative evaluations as well as a strong user evaluation that is commendable.
4. The work presents a wide amount of qualitative samples that show improvement in very common problems for diffusion models such as double heads, double hands and other issues. Samples look good.

**Weaknesses:**

1. Hard to find many weaknesses, it's a strong paper that is well written, with a clear argument, clear analysis, clear solution that simply seems to work after both quantitative and user study evaluation (and with substantial qualitative samples showing edge cases).

**Questions:**

1. I'm assuming time complexity does not vary with the new scaling factor?

**Limitations:**

1. Good limitation section. Metrics are a bit of an issue but there is a strong user eval in the paper.

---

> ### Author Rebuttal · Authors · 2023-08-10
>
> Thanks for your valuable comments. We really appreciate your interest and support in our paper.
>
> **Q1**: Relation between time complexity and the new scaling factor.
>
> **A1**: Yes, it needs constant O(1) time complexity to calculate the new scaling factor.

---

> > ### Comment · Reviewer_XLhR · 2023-08-17
> >
> > I've read the other reviews and believe their weakness comments to be minor, and relatively well addressed by the rebuttals. I will keep my initial score.

---

### Official Review · Reviewer_QLpB · 2023-07-07

**Soundness:** 3 good
**Presentation:** 4 excellent
**Contribution:** 2 fair
**Rating:** 5
**Confidence:** 5

**Summary:**

This paper analyzes the issues of using a fixed-resolution diffusion model to generate varied-size images and proposes a scaling factor to stable the attention entropy which remedies the issue. The method is evaluated on text-to-image models when the inference resolution is moderately different than the model resolution and the results show the effectiveness of the proposed method compairing to directly using fixed-resolution diffusion model for varied-size synthesis.

**Strengths:**

1. The paper is well-written and easy to follow. The motivation is clearly explained and it is with some theoretical analysis.
2. The proposed method is simple to implement, which does not require costly large scale training and could help varied-size synthesis with a fixed-resolution diffusion model.
3. The effectiveness of changing the scale factor in attention for varied-size synthesis is an interesting observation.

**Weaknesses:**

1. Synthesizing images at different resolutions is an important research problem, but it is not very clear that why a fixed-resoltion pre-trained model should be used for this task (which is the main baseline in this work). To achieve this goal, there are many other candidate methods: (1) using diffusion-based (e.g. cascaded diffusion models) or GAN-based super-resolution method to upsample the output to some resolution of powers of 2, then downsample it to the query resolution. (2) using a any-scale super-resolution network (e.g. LIIF) to upsample the network result. It is not clear that why modifying fixed-resolution model is an important research topic if it's not shown to outperform other candidate methods.
2. The test resolutions are still in a small changing range compared to the model resolution (about x0.5\~x1.5), and the FID on x0.5 scale suggests that the quality could be significantly decreased when changing the scale to be different than training scale. Does this suggest that the proposed method does not work for a wider range of scales (for example, any-scale SR works typically evaluates x1\~x4, or even extrapolates to x20\~30)?

**Questions:**

See weaknesses. It is not clear to me that, if a diffusion model only see faces at resolution 256 during training, how can its UNet synthesize a face at resolution 2048 with its kernels (trained for resolution 256) without repetitive patterns? Does the proposed method address this issue?

**Limitations:**

Yes

---

> ### Author Rebuttal · Authors · 2023-08-10
>
> Thanks for your thoughtful comments. We will explain your concerns point by point.
>
> **Q1**: Comparison to candidate methods.
>
> **A1**: We would like to point out the advantages of our method (modifying fixed-resolution models) against other methods (e.g. cascaded diffusion models and LIIF) in three key aspects.
>
> 1. **Improvement in Different Aspect Ratio**. Other methods do not support generating new images with a different aspect ratio from the original images with fixed resolutions. In comparison, our method could improve image generation in different aspect ratio, which is validated by both qualitative and quantitative experiments in our paper.
>
> 2. **Richness of Visual Information (More Important for Higher Resolutions)** . When users generate images with higher resolutions, what they are expecting is not only more pixels but also richer semantic information in images. Our method could introduce more information by enabling models to deal with more tokens, while super-resolution and other methods do not contribute to the richness of visual information by simply scaling up the original images.
>
>     To illustrate the non-trivial difference in the richness, we have provided several examples in Figure 1 in the PDF page.
>
> 3. **Time Cost and Memory Usage (More Important for Lower Resolutions)** . For diffusion models adapted by our method, their time cost and spatial usage become proportional to the generation resolution, while downsampling methods are constantly constrained to the fixed cost brought by training resolutions. In this way, our method could efficiently enables low resolution image synthesis especially on portable devices, which have a high demand for both time cost and memory usage other than image fidelity.
>
>     In Table 1, we present FID, memory usage and time cost of stable diffusion with different methods in various resolutions ($128^{2}, 224^{2}, 768^{2}, 1024^{2}$). For low resolution, our method achieves better performance in memory usage and time cost, with partial trade-off in fidelity. Noticing those fine qualitative results in Figure 3 in the paper, we believe the trade-off is acceptable considering the tackling scenario of low resolution image synthesis.
>
> | Resolution                       | GPU Memory (MiB) | FID      | Average Time (s) |
> | -------------------------------- | ---------------- | -------- | ---------------- |
> | 224 * 224 (Original)             | 7064             | 74.5742  | 5.6268           |
> | 224 * 224 (Ours)                 | 7064             | 41.8925  | 5.6274           |
> | 512 * 512 and Resize to 224      | 9324             | 21.8415  | 17.4457          |
> |                                  |                  |          |                  |
> | 768*768(Original)                | 19797            | 29.5974  | 36.7137          |
> | 768*768(Ours)                    | 19797            | 28.1372  | 36.7140          |
> | 512*512 and Resize (liif) to 768 | 9948             | 20.8712  | 18.5286          |
> |                                  |                  |          |                  |
> | 128 * 128 (Original)             | 6867             | 191.5447 | 3.4953           |
> | 128 * 128 (Ours)                 | 6867             | 127.2798 | 3.4955           |
> | 512 * 512 and Resize to 128      | 9324             | 36.0742  | 17.4377          |
> |                                  |                  |          |                  |
> | 1024 * 1024 (Original)           | 39977            | 50.5425  | 62.2414          |
> | 1024 * 1024 (Ours)               | 39978            | 43.1167  | 62.2359          |
> | 512 * 512 and Resize to 1024     | 12043            | 20.8235  | 20.8316          |
>
> Table 1: FID, memory usage and time cost of stable diffusion with different methods with different resolutions.
>
> In brief, our method does better in supporting uneven different aspect ratio and adapts diffusion models with moderate fidelity trade-off for meeting important requirements both in high resolutions and low resolutions.
>
> **Q2**: The range limit of proposed method.
>
> **A2**:  We would like to state that our method does trade off FID because we do not have the same amount of calculation as the original resolution. However, as described in A1, our method could better meet important requirements for time cost and memory usage at lower resolutions or richness of information at high resolutions. We report the results of  stable diffusion with different methods in various resolutions ($128^{2}, 224^{2}, 768^{2}, 1024^{2}$) . Additionally, our method could be implemented with no training effort to other diffusion models (e.g., diffusion models trained on the larger resolution).
>
> **Q3**: How can UNet synthesize a face at resolution 2048 with its kernels (trained for resolution 256) without repetitive patterns?
>
> **A3**: To be more specific, when the UNet is synthesizing a face at resolution 2048, our method aims to keep each token has the samilar behavior as it is at resolution 256, which means similar sized faces would be synthesized. To illustrate this, please refer to the Figure with prompt as "L'adoption internationale" in the PDF page.

---

> > ### Comment · Reviewer_QLpB · 2023-08-20
> >
> > Thank you for the detailed response. I decided to change my rating to borderline accept, but would like to highlight the following concerns:
> >
> > 1. I think the current paper may need revision to clarify the contributions of the method. To my understanding, it only enables higher-resolution synthesis of text description which have zoom-in patches observed in the training set. The issue mentioned in the abstract "higher resolution images exhibit repetitive presentation" is not really solved, for example, it still could not synthesize the prompt "a face" at an unseen high resolution, and the Figure with prompt as "L'adoption internationale" in the PDF page shows it's still repeating objects rather than generating a high-resolution version of the object.
> >
> > 2. The argument of "Richness of Visual Information" does not make sense to me. Super-resolution model is an unbiased estimation of the distribution of high-resolution images given a prompt. As a simple example, when generating "a face" at high-resolution, people would expect a 2048x2048 face with details rather than multiple faces putting together. It is misleading to claim that "richness of visual information" is an advantage of the proposed method than SR models.
> >
> > 3. Despite the weaknesses, I feel the proposed method could still be useful as it allows synthesis at higher-resolution in a reasonable range, particularly for different aspect ratios. Thus I raised my rating, but with the assumption that the paper's contribution could be clarified.

---

> > > ### Author Response · Authors · 2023-08-21
> > >
> > > Thank you for the valuable comments.
> > >
> > > 1. We have clarified the contribution by revising the issue statement as "higher resolution images exhibit repetitively disordered presentation" and imposing emphasis upon the disorder which hurts image fidelity, instead of the repetitiveness.
> > >
> > > 2. We have revised our argument to emphasize upon the difference between our method and SR, instead of the priority or the advantages, since both methods introduce more visual information from different perspectives and thus meet different needs. Additionally, they could be utilized in parallel according to users' specific requirements.
> > >
> > > We have completed the revisions and we will update the revised version after the revision submission is available.

---

### Official Review · Reviewer_wERV · 2023-07-07

**Soundness:** 3 good
**Presentation:** 2 fair
**Contribution:** 2 fair
**Rating:** 5
**Confidence:** 4

**Summary:**

This work adapts a pre-trained Stable Diffusion model for variable-resolution image generation. Since Stable Diffusion is trained on a fixed image resolution, naively varying the output size results in abnormal patterns in the images. This paper tracks the problem down to self-attention weights in the denoiser network, and identifies the so-called attention entropy as the root cause by proving that attention entropy is fundamentally correlated with image resolution. Drawing on this insight, the paper introduces resolution-dependent scaling to self-attentions to calibrate attention entropy throughout the generative process. The effectiveness of the proposed method is validated through extensive experiments.

**Strengths:**

- The method is training-free. It works by modifying the generative process of a pre-trained diffusion model at inference time.

- The method reveals a strong connection between self-attention entropy and image quality. It builds on the key finding that attention entropy is correlated with image resolution, and artifacts in low / high-resolution images can be attributed to mis-calibrated attention entropy. It thus attempts to calibrate attention entropy across various image resolutions using a resolution-dependent scaling factor. Turning theoretical insights into a simple, actionable solution is a key strength of the paper.

- The effectiveness of the method is validated by qualitative, quantitative and user study results. The improvement over the Stable Diffusion baseline seems quite consistent.

**Weaknesses:**

- A few simple baselines are missing. For example, one can simply down-sample a 512x512 image to reach a lower resolution. Similarly, one can generate a 512x512 image and subsequently up-sample it using an off-the-shelf super-resolution model. In fact, Stable Diffusion supports super-resolution and uneven aspect ratios with community effort (check out AUTOMATIC1111). To justify the main contribution of the paper (i.e., improved generation quality of variable-sized images), it is important to show that the proposed method performs equally well, or even better, compared to these simple baselines.

- Arguably, generating high-resolution images is of greater interest to the community. To this end, it would be interesting to probe the limit of the proposed method. That is, what is the highest resolution it can handle without introducing noticeable artifacts? Most high-resolution images in the paper are 768x768. There is one image in the supplement at the resolution of 1024x1024. The proposed method will generate more impact if it can further grow the image resolution.

- According to Figure 5 (left), attention entropy is up by 1 bit as image resolution grows from 512 to 768. However, Figure 6 (right) seems to suggest that applying the scaling factor only reduces entropy by a very small amount. My early impression is that the scaling factor aims to restore entropy to the level of 512x512 images. I am curious about what causes this discrepancy.

**Questions:**

Please see the section above for questions.

**Limitations:**

The paper discussed the limitations and societal impact of the method.

---

> ### Author Rebuttal · Authors · 2023-08-10
>
> Thanks for your careful and valuable comments. We will explain your concerns point by point.
>
> **Q1**: Comparison to baselines.
>
> **A1**: We would like to point out the advantages of our method (modifying fixed-resolution models) against other methods (e.g. cascaded diffusion models and LIIF) in three key aspects.
>
> 1. **Improvement in Different Aspect Ratio**. Super-resolution and downsampling methods do not support generating new images with a different aspect ratio from the original images with fixed resolutions. In comparison, our method could improve image generation in different aspect ratio, which is validated by both qualitative and quantitative experiments in our paper. For AUTOMATIC1111, the aspect ratios are passed directly into Stable Diffusion (like the original Stable Diffusion in our paper, which also support uneven aspect ratios) without any in-depth optimization.
>
> 2. **Richness of Visual Information (More Important for Higher Resolutions)** . When users generate images with higher resolutions, what they are expecting is not only more pixels but also richer semantic information in images. Our method could introduce more information by enabling models to deal with more tokens, while super-resolution and other methods do not contribute to the richness of visual information by simply scaling up the original images.
>
>     To illustrate the non-trivial difference in the richness, we have provided several examples in Figure 1 in the PDF page.
>
> 3. **Time Cost and Memory Usage (More Important for Lower Resolutions)** . For diffusion models adapted by our method, their time cost and spatial usage become proportional to the generation resolution, while downsampling methods are constantly constrained to the fixed cost brought by training resolutions. In this way, our method could efficiently enables low resolution image synthesis especially on portable devices, which have a high demand for both time cost and memory usage other than image fidelity.
>
>     In Table 1, we present FID, memory usage and time cost of stable diffusion with different methods in various resolutions ($128^{2}, 224^{2}, 768^{2}, 1024^{2}$). For low resolution, our method achieves better performance in memory usage and time cost, with partial trade-off in fidelity. Noticing those fine qualitative results in Figure 3 in the paper, we believe the trade-off is acceptable considering the tackling scenario of low resolution image synthesis.
>
> | Resolution                       | GPU Memory (MiB) | FID      | Average Time (s) |
> | -------------------------------- | ---------------- | -------- | ---------------- |
> | 224 * 224 (Original)             | 7064             | 74.5742  | 5.6268           |
> | 224 * 224 (Ours)                 | 7064             | 41.8925  | 5.6274           |
> | 512 * 512 and Resize to 224      | 9324             | 21.8415  | 17.4457          |
> |                                  |                  |          |                  |
> | 768*768(Original)                | 19797            | 29.5974  | 36.7137          |
> | 768*768(Ours)                    | 19797            | 28.1372  | 36.7140          |
> | 512*512 and Resize to 768 | 9948             | 20.8712  | 18.5286          |
> |                                  |                  |          |                  |
> | 128 * 128 (Original)             | 6867             | 191.5447 | 3.4953           |
> | 128 * 128 (Ours)                 | 6867             | 127.2798 | 3.4955           |
> | 512 * 512 and Resize to 128      | 9324             | 36.0742  | 17.4377          |
> |                                  |                  |          |                  |
> | 1024 * 1024 (Original)           | 39977            | 50.5425  | 62.2414          |
> | 1024 * 1024 (Ours)               | 39978            | 43.1167  | 62.2359          |
> | 512 * 512 and Resize to 1024     | 12043            | 20.8235  | 20.8316          |
>
> Table 1: FID, memory usage and time cost of stable diffusion with different methods with different resolutions.
>
> In brief, our method does better in supporting uneven different aspect ratio and adapts diffusion models with moderate fidelity trade-off for meeting important requirements both in high resolutions and low resolutions.
>
> **Q2**: The range limit of proposed method.
>
> **A2**: We have provided more 1024x1024 generated images in the PDF page, which show the outperforming performance and information richness of our method in synthesis. To be more specific, our method could improve FID by nearly 15% at the resolution of 1024x1024. We think the limit might depend on specific semantics (e.g., human faces) since we observe that Stable Diffusion sometimes could not successfully generate these semantics (e.g., human faces) even at the resolution of 512x512. Additionally, the proposed method could also get implemented onto other diffusion models (e.g. diffusion models trained on the resolution of 2048x2048 and get expanded to 4096x4096).
>
> **Q3**: Changing attention entropy.
>
> **A3**: We think the attention entropy should not be fixed because the information of images would naturally change in accordance with resolutions and the attention entropy would inevitably fluctuate.  Thus, the goal of our method in the paper is to "alleviate the fluctuation" instead of fixation.

---

> > ### Comment · Reviewer_wERV · 2023-08-18
> >
> > Thanks for the rebuttal. Among the three arguments to support the use case, I found the second most convincing. Please be sure to include this discussion (as well as the supporting figures) in the revision to strengthen the motivation of the work. Please also consider reporting qualitative and quantitative results of baselines combining 512x512 SD with latest SR methods (e.g., Real-ESRGAN). Lastly, as Reviewer QLpB also pointed out, growing image size by a small factor of 2 cannot sufficiently justify the merit of the method. What is the largest image size the method can support? This is an important question which is worth an ablation study. It would also be interesting to compare to MultiDiffusion mentioned by Reviewer p6J2 in this context.

---

> > > ### Author Response · Authors · 2023-08-21
> > >
> > > Thanks for the valuable comments.
> > >
> > > 1. We have included the argument as well as the supporting figures in the revision. We will update the revised version after the revision submission is available.
> > >
> > > 2. Specifically, the results denoted with *512x512 and Resize to 768/1024* shown in Table 1 in the previous rebuttal comment are the results from 512x512 SD combined with LIIF [1], which is one of the latest SR methods supporting any-scale super-resolution. Note that our method could introduce different visual information with SR methods (please refer to the PDF page). We have included the results in the revision which will be updated after the revision submission is available.
> > >
> > > 3. In Table 1 in the previous rebuttal comment, we have followed the suggestion of Reviewer QLpB to expand the evaluation range from $256^{2}$ ~ $768^{2}$ (x0.5 ~ x1.5)  to $128^{2}$ ~ $1024^{2}$ (x0.25 ~ x2), which shows that our method could effectively improve the FID for different resolutions and allow synthesis at higher-resolution in a reasonable range, which has been accepted by Reviewer QLpB.
> > >
> > > 4. We compare our method with MultiDiffusion and report corresponding FID(4K), memory usage and time cost in Table 2. As shown in Table 2, our method outperforms MultiDiffusion on each metric except for the GPU Memory with high resolution.
> > >
> > > | Resolution       | GPU Memory (MiB) $\downarrow$ | FID (4K) $\downarrow$ | Average Time (s) $\downarrow$ |
> > > | :--------------- | :---------------------------- | :-------------------- | :---------------------------- |
> > > | 224 * 224 (Ours) | 7064                          | 50.3515               | 5.6274                        |
> > > | 224 * 224 (MD)   | 7061                          | 154.6925              | 31.7680                       |
> > > | 768 * 768 (Ours) | 19797                         | 28.1372               | 36.7140                       |
> > > | 768 * 768 (MD)   | 9906                          | 40.9270               | 122.2485                      |
> > >
> > > Table 2: FID (4K samples), memory usage and time cost of diffusions with resolution $224^{2}$ and $768^{2}$.
> > >
> > > \[1]: Learning Continuous Image Representation with Local Implicit Image Function, CVPR 2021.

---

### Official Review · Reviewer_p6J2 · 2023-07-11

**Soundness:** 3 good
**Presentation:** 3 good
**Contribution:** 2 fair
**Rating:** 5
**Confidence:** 5

**Summary:**

In this paper, the authors propose a new scaling factor for attention based text-to-image generative models in order to handle variable sized generations. The authors establish the relationship between attention entropy and token size and use this newly found relationship to design a scaling factor that takes into account the image resolution. The authors also empirically show the effectiveness of this scaling factor by comparing FIDs, CLIP scores and qualitative examples.

**Strengths:**

The proposed scaling factor is very simple and easy to implement. It does not require model architecture modifications or any training of the pretrained model. The qualitative results also look very promising. Therefore I think this can be easily applied in many existing models. Overall the paper is well organized and well written.

**Weaknesses:**

1. The main formula (Equation 6 and 7) is not very well explained. There are a lot of conjectures and approximations without justifications.
2. The authors did not compare with synthesis in fixed resolution and then performing super-resolution/downsampling. Since the resolution experimented in this paper is not very far away from the pretrained resolution, it is highly likely that super-resolution/downsampling may work just as well.
3. The authors should also compare their method with other machine learning based methods such as MultiDiffusion (https://multidiffusion.github.io/)
4. The compensation for human annotator is not mentioned
5. Typo in Equation 1,3: j is used in both the numerator and the denominator of Aij

**Questions:**

1. Related to Weakness (1), Section 3.2 is still a little bit confusing to me, can the authors explain more on the connection between Equation 5, 6 and 7? I.e. How did the authors come up with Equation 6 and 7?
2. Related to Weakness (2), can the authors compare their method with synthesis in fixed pretrained resolution and perform super-resolution/downsampling? Can authors also compare to more dramatic resolution changes (eg. 512 => 128 or 512 => 1024)?
3. Related to Weakness (3), can the authors compare their method with MultiDiffusion?
4. Is FID appropriate for evaluating wrt different resolutions? Since it is trained on fixed resolution images. And how did the authors select the reference/ground truth images for FID evaluation? Did they uniformly rescale to the desired resolution or did they subsampled based on the native resolution? It may have a difference in the evaluation results.
5. Although CLIP score is a reasonable indicator, I don’t think it is a perfect metric especially when object count is involved. Have the authors considered using off-the-shelf object detectors as one of the methods to quantitatively measure the performance?

**Limitations:**

Related to Weakness (4), the compensation for human annotators is not specified in the paper.

---

> ### Author Rebuttal · Authors · 2023-08-10
>
> **Q1**: More explanation on Equation 5, 6 and 7.
>
> **A1**: In Equation 5, we derive the static relationship between attention entropy $A_{i}$ and token number $N$, i.e. $Ent(A_{i}) = \log N - \frac{1}{2} \lambda^{2} C + O(1)$, where $\lambda$ is the scaling factor and $C$ is a constant number unrelated with token number $N$. Recap that our goal is to control the fluctuating entropy $Ent(A_{i})$. Considering the form in Equation 5, we consequently set $\lambda$ with a granularity of squared logarithm (i.e., $\lambda = \alpha \sqrt{\log N} $ ), which is Equation 6 in the paper and $\alpha$ denotes a newly introduced hyper-parameter. Note that during training periods, token number $N$ is set as a constant number $T$ for training with $\lambda$ fixed as $1 / \sqrt{d}$ , which indicates that $\lambda$ comes to $1 / \sqrt{d}$  when $N = T$. In this way, we have $\lambda = \alpha \sqrt{\log T} \approx 1 / \sqrt{d}$ and we could have an approximate analytical solution for the introduced hyper-parameter $\alpha \approx 1 / (\sqrt{d\log T})$. By substituting this into Equation 6, we have $\lambda \approx \sqrt{\frac{\log_T N}{d}}$, which is Equation 7.
>
> In brief,
>
> - Considering the granularity of $\lambda$ from Equation 5, we get Equation 6.
> - Combining Equation 6 with the condition that $\lambda \approx 1 / \sqrt{d}$ when $N = T$, we get Equation 7.
>
> **Q2**: Comparison to super-resolution/downsampling methods.
>
> **A2**: We would like to point out the advantages of our method against other methods in three key aspects.
>
> 1. **Better Support in Different Aspect Ratio**. Other methods do not support generating new images with a different aspect ratio from the original images with fixed resolutions. In comparison, our method could improve image generation in different aspect ratio, which is validated by both qualitative and quantitative experiments in our paper.
>
> 2. **Richness of Visual Information (More Important for Higher Resolutions)** . When users generate images with higher resolutions, what they are expecting is not only more pixels but also richer semantic information in images. Our method could introduce more information by enabling models to deal with more tokens, while super-resolution and other methods do not contribute to the richness of visual information by simply scaling up the original images.
>
>     To illustrate the non-trivial difference in the richness, we have provided examples in Figure 1 in the PDF page.
>
> 3. **Time Cost and Memory Usage (More Important for Lower Resolutions)** . For diffusion models adapted by our method, their time cost and spatial usage become proportional to the generation resolution, while downsampling methods are constantly constrained to the fixed cost brought by training resolutions. In this way, our method could efficiently enables low resolution image synthesis especially on portable devices, which have a high demand for both time cost and memory usage other than image fidelity.
>
>     In Table 1, we present FID, memory usage and time cost of stable diffusion with different methods in resolution 224. For low resolution, our method achieves better performance in memory usage and time cost, with partial trade-off in fidelity. Noticing those fine qualitative results in Figure 3 in the paper, we believe the trade-off is acceptable considering the tackling scenario of low resolution image synthesis.
>
> | Resolution                        | GPU Memory (MiB) | FID     | Average Time (s) |
> | --------------------------------- | ---------------- | ------- | ---------------- |
> | 224 * 224 (Original)              | 7064             | 74.5742 | 5.6268           |
> | 224 * 224 (Ours)                  | 7064             | 41.8925 | 5.6274           |
> | 512 * 512 and Downsampling to 224 | 9324             | 21.8415 | 17.4457          |
>
> Table 1: FID, memory usage and time cost of stable diffusion with different methods with resolution $224^{2}$. For more results, please refer to Table 1 in other rebuttals due to character limits.
>
> **Q3**: Comparison with MultiDiffusion.
>
> **A3**: We compare our method with MultiDiffusion and report corresponding FID(4K), memory usage and time cost in Table 2. As shown in Table 2, our method outperforms MultiDiffusion in both resolutions except for the GPU Memory with high resolution.
>
> | Resolution       | GPU Memory (MiB) | FID (4K) | Average Time (s) |
> | ---------------- | ---------------- | -------- | ---------------- |
> | 224 * 224 (Ours) | 7064             | 50.3515  | 5.6274           |
> | 224 * 224 (MD)   | 7061             | 154.6925 | 31.7680          |
> | 768 * 768 (Ours) | 19797            | 28.1372  | 36.7140          |
> | 768 * 768 (MD)   | 9906             | 40.9270  | 122.2485         |
>
> Table 2: FID (4K samples), memory usage and time cost of diffusions with resolution $224^{2}$ and $768^{2}$.
>
> **Q4**: Limits on the FID metric.
>
> **A4**: In this paper we only use FID for the comparison between images with the same resolution (w/ and w/o our method) to validate the efficacy of our method. Thus, we believe the comparison is fair and credible. Specifically, we uniformly rescale the reference images to desired resolution for FID. In addition, we have discussed about the limits of FID in the Limitation Section and supported our claims with additional qualitative experiments and user study evaluation.
>
> **Q5**: Using off-the-shelf object detectors.
>
> **A5**: Please note that repetitive patterns in images are not expected to be detected by object detectors since the repetitiveness exhibits in the form of texture/composition rather than countable objects (please refer to Figure 5 in the Supplementary Materials).  Thus, we think using object detectors might not help.
>
> **Q6**: Typos and unspecified compensation.
>
> **A6**: We have revised our script by changing j in the numerator into j' and adding "For each human annotators, we pay $15 for effective completeness of user study evaluation" in Section 4.1, Line #213.

---

> > ### Comment · Reviewer_p6J2 · 2023-08-18
> > **Thank you for your response**
> >
> > Thank you for your response. I believe all my concerns have been addressed. Therefore I would like to raise my score from 4 to 5.

---

### Author Rebuttal · Authors · 2023-08-10

Thanks for all the reviewers and AC for your time and valuable comments!

This comment is followed by the PDF page with new figures to support our views in information richness.

---

### Decision · Program_Chairs · 2023-09-21

**Decision:**

Accept (poster)

**Comment:**

All the reviewers lean towards the acceptance of the work. Reviewers appreciated the proposed training-free technique for variable size image generation as well as the technical insights. Reviewers raised several clarification questions and also raised concerns regarding some missing comparisons. Authors addressed several of these questions in their responses. The reviewers did raise some valuable concerns and suggestions that should be addressed in the final camera-ready version of the paper, which include adding the relevant rebuttal discussions and revisions in the main paper. The authors are encouraged to make the necessary changes to the best of their ability.